# Mamba-3: Improved Sequence Modeling using State Space Principles

**Aakash Lahoti**[*1]    **Kevin Y. Li**[*1]    **Berlin Chen**[*2]    **Caitlin Wang**[*2]    **Aviv Bick**[1]
**J. Zico Kolter**[1]    **Tri Dao**[2,3]    **Albert Gu**[1,4]
[1]Carnegie Mellon University    [2]Princeton University    [3]Together AI    [4]Cartesia AI
{alahoti, kyl2, abick, zkolter, agu}@cs.cmu.edu
{bc2188, caitlinwang, tridao}@princeton.edu

## ABSTRACT

Scaling inference-time compute has emerged as an important driver of LLM performance, making inference efficiency a central focus of model design alongside model quality. While the current Transformer models deliver strong model quality, their quadratic compute and linear memory makes inference expensive. This has spurred the development of sub-quadratic models with reduced linear compute and constant memory requirements. However, many recent linear models trade off model quality and capability for algorithmic efficiency, failing on tasks such as state tracking. Moreover, their theoretically linear inference remains hardware-inefficient in practice. Guided by an inference-first perspective, we introduce three core methodological improvements inspired by the state space model (SSM) viewpoint of linear models. We combine: (1) a more expressive recurrence derived from SSM discretization, (2) a complex-valued state update rule that enables richer state tracking, and (3) a multi-input, multi-output (MIMO) formulation for better model performance without increasing decode latency. Together with architectural refinements, our **Mamba-3** model achieves significant gains across retrieval, state-tracking, and downstream language modeling tasks. At the 1.5B scale, Mamba-3 improves average downstream accuracy by 0.6 percentage points compared to the next best model (Gated DeltaNet), with Mamba-3's MIMO variant further improving accuracy by another 1.2 points for a total 1.8 point gain. Across state-size experiments, Mamba-3 achieves comparable perplexity to Mamba-2 despite using half of its predecessor's state size. Our evaluations demonstrate Mamba-3's ability to advance the performance-efficiency frontier.

## 1 INTRODUCTION

Test-time compute has emerged as a key driver of progress in LLMs, with techniques like chain-of-thought reasoning and iterative refinement demonstrating that inference-time scaling can unlock new capabilities (Wu et al., 2025; Snell et al., 2024). The rapid rise of parallel, agentic workflows has only intensified the need for efficient inference and deployment of such models (OpenAI, 2026; Anthropic, 2026). This paradigm shift makes inference efficiency (Kwon et al., 2023; Li et al., 2024) paramount, as the impact of AI systems depends critically on their ability to perform large-scale inference during deployment. Model architecture design plays a fundamental role in determining inference efficiency, as architectural choices directly dictate the computational and memory requirements during generation. While Transformer-based models (Vaswani et al., 2017) are the current industry standard, they are fundamentally bottlenecked by linearly increasing memory demands through the KV cache and quadratically increasing compute requirements through the self-attention mechanism. These drawbacks have motivated recent lines of work on sub-quadratic models, e.g., state space models (SSMs) and linear attention, which retain constant memory and linear compute while attaining comparable or better performance than their Transformer counterparts. These models have achieved widespread adoption, with layers such as Mamba-2 (Dao & Gu, 2024) and Gated DeltaNet (GDN) (Schlag et al., 2021; Yang et al., 2025d) incorporated into large-scale hybrid models that match the performance of pure Transformer alternatives with much higher efficiency (NVIDIA et al., 2025; Yang et al., 2025a; Kimi Team et al., 2025; Tencent Hunyuan Team et al., 2025).

Despite the success of linear models, significant progress remains in improving their performance, in particular on advancing the Pareto frontier between model quality and inference efficiency. For example, Mamba-2 was developed to improve training speed over Mamba-1 (Gu & Dao, 2024), by sacrificing some expressivity

---

[*]Equal contribution.

and thus performing worse for inference-matched models. In addition, they have been shown to lack certain capabilities, such as poor state-tracking abilities, e.g., simply determining parity of bit sequences (Grazzi et al., 2025; Sarrof et al., 2024). Finally, despite these sub-quadratic models being prized for theoretically efficient inference and thus their widespread adoption, their inference algorithms are not hardware efficient. In particular, because these algorithms were developed from a training perspective, their decoding phase has low arithmetic intensity (the ratio of FLOPs to memory traffic), resulting in large portions of hardware remaining idle.

To develop more performant models from an inference-first paradigm, we introduce three core SSM-centric methodological changes to Mamba-2.

**Exponential-Trapezoidal Discretization.**    We provide a simple technique for discretizing time-varying, selective SSMs. Through our framework, we can derive several new discretization methods. One of our instantiations, referred to as "exponential-Euler," formalizes Mamba-1 and Mamba-2's heuristic discretization that previously lacked theoretical justification. Our new "exponential-trapezoidal" instantiation is a more expressive, generalization of "exponential-Euler," where the recurrence can be expanded to reveal an implicit convolution applied on the SSM input. Combined with explicit $B, C$ biases, Mamba-3 can empirically replace the short causal convolution in language model architectures, which was previously hypothesized to be essential for recurrent models.

**Complex-valued State Space Model.**    By viewing the underlying SSM of Mamba-3 as complex-valued, we enable a more expressive state update than Mamba-2's. This change in update rule, designed to be lightweight for training and inference, overcomes the lack of state-tracking ability common in many current linear models. Our complex-valued update rule is equivalent to a data-dependent rotary embedding and can be efficiently computed (Su et al., 2023), and we empirically demonstrate its ability to solve synthetic tasks outside the capabilities of prior linear models.

**Multi-Input, Multi-Output (MIMO) SSM.**    To improve FLOP efficiency during decoding, we switch from an outer-product–based state update to a matrix-multiplication–based state update. From the view of the signal processing foundations of SSMs, such a transition exactly coincides with the generalization from a single-input single-output (SISO) sequence dynamic to a multiple-input multiple-output (MIMO) one. Here, we find that MIMO is particularly suitable for inference, as the extra expressivity enables more computation during the memory-bound state update during decoding, without increasing the state size and compromising speed.

Put together, these improvements form the core of our **Mamba-3** layer. Methodologically, we note that these all arise naturally from an SSM-centric perspective but are not immediate from other popular viewpoints of modern linear layers such as linear attention or test-time regression; we discuss these connections further in Section A. Empirically, we validate our new model's abilities and capabilities on a suite of synthetic state-tracking and language-modeling tasks.

- **Better Quality.** At 1.5B scale, Mamba-3 (MIMO) improves average downstream language modeling accuracy by **+2.2** over Transformers, **+1.9 points** over Mamba-2, and **+1.8** over GDN. Mamba-3 (SISO) improves over the next best model, GDN, by **+0.6** points. Furthermore, across state size experiments, Mamba-3 (MIMO) with state size 64 matches the perplexity of Mamba-2 with state size 128, effectively achieving the **same language modeling performance with half the latency**.

- **New Capabilities.**    Mamba-3's complexification of the SSM state enables it to **solve synthetic state-tracking tasks that Mamba-2 cannot**. We empirically demonstrate that the efficient RoPE-like calculation is able to near perfectly solve arithmetic tasks, while Mamba-3 without RoPE and Mamba-2 perform not better than random guessing.

- **Inference Efficiency.** Mamba-3 (MIMO) improves hardware utilization. It increases decoding FLOPs by up to **4×** relative to Mamba-2 at fixed state size, while maintaining **similar wall-clock decode latency**, and simultaneously improving perplexity and downstream performance. We release fast training and inference kernels for Mamba-3.

Mamba-3 (SISO) improves quality and capability over prior linear models, and Mamba-3 (MIMO) further improves performance over Mamba-3 (SISO) and other strong baselines while matching inference speed with Mamba-2. Both of our Mamba-3 variants advance the performance-latency Pareto frontier through their strong modeling capabilities and hardware-efficient design.

## 2 PRELIMINARIES

### 2.1 NOTATION

Scalars are denoted by plain-text letters (e.g., $x, y$). Tensors, including vectors and matrices, are denoted by bold letters (e.g., $\boldsymbol{h}, \boldsymbol{C}$). The shape of the tensor can be inferred from the context. We denote the input sequence length as $T$, the model dimension as $D$, and the SSM state size as $N$. For time indices, we use subscripts (e.g., $x_t$ for the input at time $t$). The Hadamard product between two tensors is denoted by $\odot$. For a vector of size $\boldsymbol{v} \in \mathbb{R}^d$, we denote $\mathrm{Diag}(\boldsymbol{v}) \in \mathbb{R}^{d \times d}$ as the diagonal matrix with the vector $\boldsymbol{v}$ as the diagonal, and for products of scalars across time steps, we use the notation $\alpha_{t \cdots s} = \alpha_{t:s}^{\times} = \prod_{i=s}^{t} \alpha_i$.

### 2.2 SSM PRELIMINARIES

State Space Models (SSMs) describe continuous-time linear dynamics via

$$\dot{\boldsymbol{h}}(t) = \boldsymbol{A}(t)\boldsymbol{h}(t) + \boldsymbol{B}(t)x(t), \qquad\qquad y(t) = \boldsymbol{C}(t)^{\top}\boldsymbol{h}(t),$$

where $\boldsymbol{h}(t) \in \mathbb{R}^N$ is the hidden state, $x(t) \in \mathbb{R}$ the input, and $\boldsymbol{A}(t) \in \mathbb{R}^{N \times N}$, $\boldsymbol{B}(t), \boldsymbol{C}(t) \in \mathbb{R}^N$. We will occasionally refer to $\boldsymbol{A}(t)$ as the *state-transition* and $\boldsymbol{B}(t)x(t)$ as the *state-input*; this also extends to their discretized counterparts. For discrete sequences with step size $\Delta_t$, Mamba-1 and Mamba-2 *discretized* the system to the following recurrence

$$\boldsymbol{h}_t = e^{\Delta_t \boldsymbol{A}_t}\boldsymbol{h}_{t-1} + \Delta_t \boldsymbol{B}_t x_t, \qquad\qquad y_t = \boldsymbol{C}_t^{\top}\boldsymbol{h}_t.$$

**Mamba-2's Parameterization.** The core of the Mamba-2 layer (Dao & Gu, 2024) is a *data-dependent* and hardware-efficient SSM. Both the state-transition and state-input are made data-dependent through the projection of $\Delta_t \in \mathbb{R}_{>0}$ and $\boldsymbol{B}, \boldsymbol{C} \in \mathbb{R}^N$ from the current token. By parameterizing the state-transition $\boldsymbol{A}_t$ as a scalar times identity, the SSM recurrence can be efficiently computed with the matrix multiplication tensor cores of GPUs. Defining $\alpha_t := e^{\Delta_t A_t} \in (0, 1)$ and $\gamma_t := \Delta_t$, the update becomes

$$\boldsymbol{h}_t = \alpha_t \boldsymbol{h}_{t-1} + \gamma_t \boldsymbol{B}_t x_t, \qquad y_t = \boldsymbol{C}_t^{\top}\boldsymbol{h}_t. \tag{1}$$

The data-dependent state-transition $\alpha_t$ controls the memory horizon of each SSM within the layer. $\Delta_t$ in particular modulates both the state-transition and state-input: a larger $\Delta_t$ forgets faster and up-weights the current token, while a smaller $\Delta_t$ retains the hidden state with minimal contributions from the current token.

*Remark* 1. Mamba-3 switches to data-dependent $A_t$ to keep all SSM parameters data-dependent. This change does not change the empirical performance of the model.

### 2.3 STRUCTURED MASKED REPRESENTATION AND STATE SPACE DUALITY

Mamba-2 showed that a large class of SSMs admit a *matrix* form that vectorizes the time-step recurrence. Through the state space duality (SSD) framework, recurrent SSMs can be represented within a parallel form that incorporates an element-wise mask to model the state-transition decay. SSD provides a general framework for a duality between linear recurrence and parallelizable (matrix-multiplication-based) computational forms

$$\boldsymbol{Y} = (\boldsymbol{L} \odot \boldsymbol{C}\boldsymbol{B}^{\top})\boldsymbol{X} \tag{2}$$

where $\boldsymbol{L} \in \mathbb{R}^{T \times T}$ is a structured mask, $\boldsymbol{B}, \boldsymbol{C} \in \mathbb{R}^{T \times N}$, $\boldsymbol{X} \in \mathbb{R}^{T \times D}$ are the inputs to the SSM and $\boldsymbol{Y} \in \mathbb{R}^{T \times D}$ is its output. Different structures on $\boldsymbol{L}$ give rise to various instantiations of SSD. Equation (2) also draws a general connection between recurrence and attention, by setting $\boldsymbol{Q} := \boldsymbol{C}$, $\boldsymbol{K} := \boldsymbol{B}$, $\boldsymbol{V} := \boldsymbol{X}$ and viewing $\boldsymbol{L}$ as a data-dependent mask. In fact, the simplest case of SSD is (causal) linear attention (Katharopoulos et al., 2020), where $\boldsymbol{L}$ is the causal triangular mask. Mamba-2 is a generalization where $\boldsymbol{L}_{ij} = 0$ for $i < j$, $\boldsymbol{L}_{ij} = \alpha_{i \ldots j}\gamma_j$ for $i \geq j$, where the terms $\alpha_t, \gamma_t$ are from equation (1) and $\alpha_0 = 1$.[1] In Section B.3.1, we show that Mamba-3 is a generalization of Mamba-2 with a more expressive $\boldsymbol{L}$, and hence also an instance of SSD.

## 3 METHODOLOGY

We introduce Mamba-3, with three innovations rooted in classical state space theory—"exponential-trapezoidal" discretization for more expressive dynamics, complex-valued SSM for state tracking, and multi-input multi-output (MIMO) for improved performance and hardware utilization for inference—to address the quality, capability, and efficiency issues of current linear models.

### 3.1 EXPONENTIAL-TRAPEZOIDAL DISCRETIZATION

Structured SSMs are naturally defined as continuous-time dynamical systems that map input functions, $x(t) \in \mathbb{R}$, to output functions, $y(t) \in \mathbb{R}$, for time $t > 0$. The underlying system is a first-order ordinary differential equation (ODE) for the state $\dot{\boldsymbol{h}}(t)$ and a dot-product readout for the output $y(t)$. In sequence

---

[1] In this paper, $\boldsymbol{B}_t$ represents the continuous parameter, whereas in Mamba-2, $\boldsymbol{B}_t$ represents the discretized parameter which is equivalent to $\gamma_t \boldsymbol{B}_t$, folding in the $\gamma$ term.

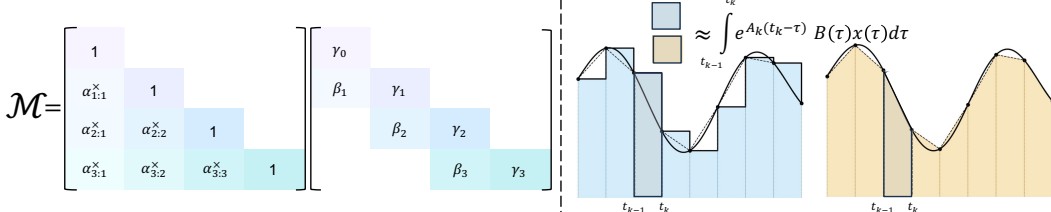

Figure 1: **Left:** The structured mask induced by the exponential-trapezoidal rule (Section 3.1) is a product of the decay and two-band convolutional mask. **Right:** Euler (hold endpoint) versus Trapezoidal (average endpoints) integral approximation.

modeling, since the data is observed at discrete time steps, it requires applying a *discretization step* to the SSM to transform its continuous-time dynamics into a discrete recurrence.

Discretization methods are well-studied in classical control theory with several canonical formulas used in earlier SSM works in deep learning (Gu et al., 2022a;b; Smith et al., 2023). These mechanisms were traditionally stated and applied to linear-time invariant (LTI) systems such as S4, and their derivations do not directly apply to linear-time varying (LTV) systems such as Mamba-1/2. Additionally, while Mamba-1 adapted the zero-order hold (ZOH) method to LTV systems, the complexity associated with selective SSMs prompted the use of heuristic approximations without theoretical justification and did not correspond to any established discretization technique. In the following subsection, we formalize the previous heuristics used in current LTV SSMs through our discretization framework and utilize it to propose a more expressive discretization scheme.

### 3.1.1 OVERVIEW OF EXPONENTIAL-ADJUSTED DISCRETIZATION

We introduce a simple derivation that leads to a class of new discretization methods for LTV state space models. The method can be instantiated in various ways; we show that one instantiation results in the heuristic used in Mamba-1/2, thereby theoretically justifying it (exponential-Euler). We also introduce a more powerful discretization (exponential-trapezoidal) used in Mamba-3.

We begin with the observation that the closed form solution of the simple linear ODE, $\boldsymbol{h}'(t) = A\boldsymbol{h}(t)$, is $\boldsymbol{h}(t) = e^{tA}\boldsymbol{h}(0)$, where the dynamics are dominated by an exponential scaling. We extend this analysis to the "adjusted system" that features a time-dependent transition scalar $A(t)$ as well as a system input $\boldsymbol{B}(t)x(t)$. Solving this,

$$\boldsymbol{h}(\tau_t) = \underbrace{\exp\left(\int_{\tau_{t-1}}^{\tau_t} A(s)ds\right)\boldsymbol{h}(\tau_{t-1})}_{\text{solved analytically}} + \underbrace{\int_{\tau_{t-1}}^{\tau_t} \exp\left(\int_{\tau}^{\tau_t} A(s)ds\right)\boldsymbol{B}(\tau)x(\tau)d\tau}_{\text{approximated}}$$

We approximate the first integral by using $A(s) := A(\tau_t)$, for $s \in [\tau_{t-1}, \tau_t]$,

$$\boldsymbol{h}_t \approx \exp(\Delta_t A_t)\boldsymbol{h}_{t-1} + \int_{\tau_{t-1}}^{\tau_t} \exp((\tau_t - \tau)A_t)\boldsymbol{B}(\tau)x(\tau)d\tau$$

which serves as the foundation for further discretization techniques in a TV setting. The full derivation is detailed in Proposition 5. We recover the prior Mamba discretizations from our framework in Section B.1.

**Exponential-Trapezoidal (Mamba-3).** However, Euler's rule from previous Mambas provides only a first-order approximation of the state-input integral: the local truncation error is $O(\Delta_t^2)$, which accumulates across steps to yield a global error of $O(\Delta_t)$ over the sequence. In contrast, we introduce a *generalized trapezoidal rule*, which provides a second-order accurate approximation of the integral, offering improved accuracy over Euler's rule. Specifically, it approximates the integral with a *data-dependent, convex combination of both interval endpoints*. This generalization extends the classical trapezoidal rule (Süli & Mayers, 2003), which simply averages the interval endpoints (Figure 1).

**Proposition 1** (Generalized Trapezoidal Recurrence). *Approximating the state-input integral in Equation (10) by the general trapezoidal rule yields the recurrence,*

$$\boldsymbol{h}_t = e^{\Delta_t A_t}\boldsymbol{h}_{t-1} + (1-\lambda_t)\Delta_t e^{\Delta_t A_t}\boldsymbol{B}_{t-1}x_{t-1} + \lambda_t\Delta_t\boldsymbol{B}_t x_t, \tag{3}$$

$$:= \alpha_t\boldsymbol{h}_{t-1} + \beta_t\boldsymbol{B}_{t-1}x_{t-1} + \gamma_t\boldsymbol{B}_t x_t, \tag{4}$$

*where $\lambda_t \in [0,1]$ is a data-dependent scalar, $\alpha_t := e^{\Delta_t A_t}$, $\beta_t := (1-\lambda_t)\Delta_t e^{\Delta_t A_t}$, $\gamma_t := \lambda_t\Delta_t$.*

---

[1]The actual Mamba implementation follows https://github.com/state-spaces/mamba/issues/129.

*Remark* 2 (Expressivity). The exponential-trapezoidal rule is a generalization of (a) the classical trapezoid rule, recovered when $\lambda_t = \frac{1}{2}$, and (b) Mamba-2's Euler's rule, recovered when $\lambda_t = 1$.

*Remark* 3 (Error Rate). This is a second-order discretization with local truncation error $O(\Delta_t^3)$ and global error $O(\Delta_t^2)$ over the sequence under standard stability assumptions, provided that the trapezoidal parameter satisfies $\lambda_t = \frac{1}{2} + O(\Delta_t)$. However, our ablations indicate that not enforcing this constraint is the best for empirical performance. See Appendix B.3,B.4 for details.

Our discretization framework and its two instantiations, exponential-Euler and exponential-trapezoidal, are, to the best of our knowledge, novel for structured SSMs in deep learning. Table 5 compares and summarizes the canonical and common discretization schemes for SSMs.

### 3.1.2 EXPONENTIAL-TRAPEZOIDAL RECURRENCE AS AN IMPLICIT CONVOLUTION

Our generalized exponential-trapezoidal discretization is equivalent to applying a *data-dependent* convolution of size two on the state-input. A normal recurrent SSM materializes the state-input $v_t = B_t x_t$, then computes a linear recurrence $h_t = \alpha_t h_{t-1} + \gamma_t v_t$. Now, we instead first apply a width-2 convolution on $v_t$ (weighted by $\beta, \gamma$) before passing it into the linear recurrence (4).

*Remark* 4 (Convolution Differences). There is a distinct difference between the "convolution" induced by exponential-trapezoidal discretization, and the standard short convolutions used by sequence models such as Mamba and Gated DeltaNet. Standard short convolutions are independent operations applied on $x_t$ (and often $B_t, C_t$) *outside* the core recurrence, while our new discretization can be interpreted as a convolution on the *state-input* $B_t x_t$ *within* the core recurrence.

We also show that the exponential-trapezoidal discretization can be viewed as a convolution applied to the mask matrix $L$ in the SMA viewpoint (Appendix B.3.1)

### 3.2 COMPLEX-VALUED SSMS

Modern SSMs are designed with efficiency as the central goal, motivated by the need to scale to larger models and longer sequences. For instance, successive architectures have progressively simplified the state transition matrix: S4 (Gu et al., 2022a) used complex-valued Normal Plus Low Rank (NPLR) matrices, Mamba (Gu & Dao, 2024) reduced this to a diagonal of reals, and Mamba-2 (Dao & Gu, 2024) further simplified it to a single scaled identity. Although these simplifications largely maintain language modeling performance, recent works (Merrill et al., 2025; Sarrof et al., 2024; Grazzi et al., 2025) have shown that the restriction to real, non-negative eigenvalue transitions degrades the capabilities of the model on simple state-tracking tasks. This limitation, formalized in Theorem 1 of (Grazzi et al., 2024), arises from restricting the eigenvalues of the transition matrix to real numbers, which cannot represent "rotational" hidden state dynamics. Parity, the function on binary inputs $\{0,1\}$, defined as $\sum_t x_t \mod 2$, is one such task. It can be performed using update: $h_t = R(\pi x_t) h_{t-1}$, where $R(\cdot)$ is a 2-D rotation matrix. Such rotational dynamics cannot be expressed with real eigenvalues.

### 3.2.1 COMPLEX SSM WITH EXPONENTIAL-EULER DISCRETIZATION

To recover this capability, we begin with complex SSMs (5), which *are* capable of representing state-tracking dynamics. We show that, under discretization (Proposition 5), complex SSMs can be formulated as a real SSMs with a *block-diagonal transition matrix composed of* $2 \times 2$ *rotation matrices* (Proposition 2). We then show that this is equivalent to applying *data-dependent rotary embeddings* on both the input and output projections $B, C$ respectively , which establishes a theoretical connection between complex SSMs and data-dependent RoPE embeddings (Proposition 3). Finally, the "RoPE trick" used in Su et al. (2023) allows for an efficient implementation complex-valued state-transition matrices with minimal computational overhead compared to real-valued SSMs.

**Proposition 2** (Complex-to-Real SSM Equivalence). *Consider a complex-valued SSM*

$$\dot{h}(t) = \text{Diag}\big(A(t) + i\theta(t)\big) h(t) + \big(B(t) + i\hat{B}(t)\big) x(t), \tag{5}$$

$$y(t) = \text{Re}\Big(\big(C(t) + i\hat{C}(t)\big)^\top h(t)\Big),$$

*where* $h(t) \in \mathbb{C}^{N/2}$, $\theta(t), B(t), \hat{B}(t), C(t), \hat{C}(t) \in \mathbb{R}^{N/2}$, *and* $x(t), A(t) \in \mathbb{R}$. *Under exponential-Euler discretization, this system is equivalent to a real-valued SSM*

$$h_t = e^{\Delta_t A_t} R_t h_{t-1} + \Delta_t B_t x_t, \tag{6}$$

$$y_t = C_t^\top h_t,$$

with state $\boldsymbol{h}_t \in \mathbb{R}^N$, projections

$$\boldsymbol{B}_t := \begin{bmatrix} \boldsymbol{B}_t \\ \hat{\boldsymbol{B}}_t \end{bmatrix} \in \mathbb{R}^N, \qquad \boldsymbol{C}_t := \begin{bmatrix} \boldsymbol{C}_t \\ -\hat{\boldsymbol{C}}_t \end{bmatrix} \in \mathbb{R}^N,$$

and a transition matrix

$$\boldsymbol{R}_t := Block\left(\{R(\Delta_t \boldsymbol{\theta_t}[i])\}_{i=1}^{N/2}\right) \in \mathbb{R}^{N \times N}, \qquad R(\theta) := \begin{bmatrix} \cos(\theta) & -\sin(\theta) \\ \sin(\theta) & \cos(\theta) \end{bmatrix}.$$

The proof is given in Section C.1.

Proposition 2 shows that the discretized complex SSM of state dimension $N/2$ has an equivalent real SSM with doubled state dimension ($N$), and its transition matrix is a scalar decayed block-diagonal matrix of $2 \times 2$ data-dependent rotation matrices ($e^{\Delta_t A_t} \boldsymbol{R}_t$).

**Proposition 3** (Complex SSM, Data-Dependent RoPE Equivalence). *Under the notation established in Proposition 2, consider the real SSM defined in equation (6) unrolled for $T$ time-steps. The output of the above SSM is equivalent to that of a vanilla scalar transition matrix-based SSM (11) with a data-dependent rotary embedding applied on the $\boldsymbol{B}, \boldsymbol{C}$ components of the SSM, as defined by:*

$$\boldsymbol{h}_t = e^{\Delta_t A_t} \boldsymbol{h}_{t-1} + \left(\prod_{i=0}^{t} \boldsymbol{R}_i^\top\right) \boldsymbol{B}_t x_t, \qquad y_t = \left[\left(\prod_{i=0}^{t} \boldsymbol{R}_i^\top\right) \boldsymbol{C}_t\right]^\top \boldsymbol{h}_t \tag{7}$$

*where the matrix product represents right matrix multiplication, e.g., $\prod_{i=0}^{1} \boldsymbol{R}_i = \boldsymbol{R}_0 \boldsymbol{R}_1$. We refer to the usage of a transformed real-valued SSM to compute the complex SSM as the "RoPE trick."*

The proof is given in Section C.2.

To observe the connection of complex SSMs to RoPE embeddings, note that in the above proposition, the data-dependent rotations $\boldsymbol{R}_i$ are aggregated across time-steps and applied to $\boldsymbol{C}, \boldsymbol{B}$, which, by the state space duality framework, correspond to the query ($\boldsymbol{Q}$) and key ($\boldsymbol{K}$) components of attention (Section 2.3). Analogously, vanilla RoPE (Su et al., 2023) applies *data-independent* rotation matrices, where the rotation angles follow a fixed frequency schedule $\boldsymbol{\theta}[i] = 10000^{-2i/N}$.

### 3.2.2 COMPLEX SSM WITH EXPONENTIAL-TRAPEZOIDAL DISCRETIZATION

After deriving the recurrence for complex SSMs with exponential-Euler discretization, the generalization to exponential-trapezoidal discretization is similar. Proposition 4 provides the full recurrence with the RoPE trick for Mamba-3.

**Proposition 4** (Rotary Embedding Equivalence with Exponential-Trapezoidal Discretization). *Discretizing a complex SSM with the exponential-trapezoidal rule (Proposition 1) yields the recurrence*

$$\boldsymbol{h}_t = \alpha_t \boldsymbol{h}_{t-1} + \beta_t \left(\prod_{i=0}^{t-1} \boldsymbol{R}_i^\top\right) \boldsymbol{B}_{t-1} x_{t-1} + \gamma_t \left(\prod_{i=0}^{t} \boldsymbol{R}_i^\top\right) \boldsymbol{B}_t x_t,$$

$$y_t = \left[\left(\prod_{i=0}^{t} \boldsymbol{R}_i^\top\right) \boldsymbol{C}_t\right]^\top \boldsymbol{h}_t. \tag{8}$$

*Here $\boldsymbol{R}_t$ is the block-diagonal rotation matrix defined in Proposition 3.*

The proof is in Section C.3.

We empirically validate that our complex SSM, implemented via data-dependent RoPE, is capable of solving state-tracking tasks that real-valued SSMs with and without standard RoPE cannot (Table 3b), supporting theoretical claims.

### 3.3 MULTI-INPUT, MULTI-OUTPUT

Scaling test-time compute has opened new frontiers in model capability, such as agentic workflows, where inference takes up an increasing share of the overall compute budget. This has placed a renewed focus on inference efficiency of language models and spurred the adoption of SSMs and sub-quadratic layers which feature fixed-sized hidden states and thus offer lower compute and memory requirements. Although these new layers have a lower wall-clock time compared to Transformers, their decoding is heavily memory-bound, resulting in low hardware utilization. In this section, we use the SSM perspective to introduce a methodological refinement to the Mamba-3 recurrence that allows for *increased model FLOPs without increasing decoding wall-clock time, resulting in a better model with the same decoding speed.*

**Decoding Arithmetic Intensity.** To improve hardware efficiency, we need to consider the arithmetic intensity of token generation, defined as FLOPs divided by the number of input-output bytes for a given op. Since SSM decoding saturates the memory bandwidth with idle compute (i.e., being *memory-bound*), we would like to increase its arithmetic intensity to effectively overlay compute with memory I/O. More concretely, the arithmetic intensity for a single generation in Mamba is around 2.5 ops per byte (Table 7a), while the arithmetic intensity for bfloat16 matmul is about 295 ops per byte for NVIDIA H100-SXM5 (NVIDIA, 2022). Consequently, SSM decoding falls far short of a compute-bound regime, and moreover it is not clear how one can adjust the existing parameters in Mamba to mitigate the lack of hardware efficiency. We note that this observation applies generally to other sub-quadratic models, such as causal linear attention.

**From SISO to MIMO.** Consider a single head of a typical SSM with *head dimension* $P$, which involves stacking the SISO recurrence $\boldsymbol{h}_t \leftarrow \alpha_t \boldsymbol{h}_{t-1} + \boldsymbol{B}_t x_t$ with $P$ copies sharing the same $\alpha_t$ and $\boldsymbol{B}_t$. The resulting broadcasted recurrence $\boldsymbol{h}_t \leftarrow \alpha_t \boldsymbol{h}_{t-1} + \boldsymbol{B}_t \boldsymbol{x}_t^\top$ takes inputs $\boldsymbol{x}_t \in \mathbb{R}^P$ and produces states $\boldsymbol{h}_t \in \mathbb{R}^{N \times P}$.

Note that the memory traffic (input/output size) is dominated by the state $\boldsymbol{h}_t$, while the computation mainly comprises the outer product $\boldsymbol{B}_t \boldsymbol{x}_t^\top$ which has FLOPs proportional to $NP$. We observe that by increasing the dimension of the latter terms, transforming $\boldsymbol{B}_t \in \mathbb{R}^N \to \boldsymbol{B}_t \in \mathbb{R}^{N \times R}$ and $\boldsymbol{x}_t \in \mathbb{R}^P \to \boldsymbol{x}_t \in \mathbb{R}^{P \times R}$, the memory traffic does not significantly increase (for small $R$) while the FLOPs consumed increases by a factor of $R$ (Table 7a). Thus, this transformation increases the arithmetic intensity of the recurrence. Furthermore, the increased arithmetic intensity is translated into practical gains because the outer product $\boldsymbol{B}_t \boldsymbol{x}_t^\top$ becomes a hardware-efficient matrix-matrix product (matmul) which incurs very little overhead. As a result, the new recurrence is more expressive than the original while practically preserving the decoding speed.

For similar reasons, the computation of the output from the state, $\boldsymbol{y}_t \leftarrow \boldsymbol{C}_t^\top \boldsymbol{h}_t$ acquires an extra rank $R$, by transforming the output projection as $\boldsymbol{C}_t \in \mathbb{R}^N \to \boldsymbol{C}_t \in \mathbb{R}^{N \times R}$. Overall, this transformation is equivalent to expanding the original single-input, single-output (SISO) recurrence to multi-input, multi-output (MIMO). We note that MIMO SSMs incur a slow-down during training due to an $R$-fold increase in FLOPs with more compute-bound kernels, even though the decoding wall-clock performance is not significantly impacted due to its memory-bound kernels. Further details on the usage of MIMO SSMs in Mamba-3 are in Section D.

## 3.4 MAMBA-3 ARCHITECTURE

The overall architecture follows Llama (Grattafiori et al., 2024), alternating Mamba-3 and SwiGLU blocks with pre-norm. The Mamba-3 block retains the overall layout of its predecessor, while introducing several key modifications.

**Updated SSM Recurrence.** The SSD layer is replaced with the more expressive complex-valued exponential-trapezoidal SSM defined in Proposition 4. Mamba-3 employs the SISO SSM by default to enable fair comparisons with other SISO-like models, but its MIMO variant can be trained and deployed as a stronger alternative to baseline Mamba-3 (Table 1).

**BC / QK Normalization.** RMS normalizations are added following the $\boldsymbol{B}, \boldsymbol{C}$ projection, mirroring the QKNorm commonly used in modern Transformers (Henry et al., 2020; Wortsman et al., 2023) and other recent linear models (Yang et al., 2025c; Hu et al., 2025). We call this either BCNorm or QKNorm interchangeably. We find that BCNorm is also able to stabilize large-scale runs, resulting in the removal of the post-gate RMSNorm layer (introduced in Mamba-2 for stability) in our pure Mamba-3 models. However, in hybrid models, the removed RMSNorm is crucial for long-context extrapolation (Table 2).

$\boldsymbol{B}, \boldsymbol{C}$ **Biases.** Similarly to Yu & Erichson (2025), which proved adding channel-specific bias to $\boldsymbol{B}$ in a blockwise variant of Mamba-1 grants universal approximation capabilities, Mamba-3 incorporates learnable, head-specific, channel-wise biases into the $\boldsymbol{B}$ and $\boldsymbol{C}$ after BCNorm. We hypothesize that these biases also endow the model with more convolution-like behavior, where the added biases in $\boldsymbol{B}, \boldsymbol{C}$ create data-independent SSMs that function more similarly to convolutions. Ablations on the bias parameterization are located in Section G). The combination of data-independent bias parameters, together with exponential-trapezoidal discretization (which itself induces a convolution on the state-input), is empirically able to obviate the short causal convolution and its accompanying activation function present in Mamba-2 and most modern recurrent models (Section 4.2).

## 4 EMPIRICAL VALIDATION

### 4.1 LANGUAGE MODELING

All models are pretrained with 100B tokens of the FineWeb-Edu dataset (Penedo et al., 2024) with the Llama-3.1 tokenizer (Grattafiori et al., 2024) at a 2K context length with the same standard training protocol. Training and evaluation details can be found in Section E.

Table 1: Downstream language modeling evaluations on models trained with 100B FineWeb-Edu tokens. Best results are **bolded**, and second best are underlined, excluding Mamba-3 MIMO variants. All models are trained with the same procedure. Mamba-3 SISO outperforms Mamba-2 and others at every model scale, and MIMO with rank $R=4$ further improves modeling capabilities.

| Model | FW-Edu ppl ↓ | LAMB. ppl ↓ | LAMB. acc ↑ | HellaS. acc_n ↑ | PIQA acc ↑ | Arc-E acc ↑ | Arc-C acc_n ↑ | WinoGr. acc ↑ | OBQA acc ↑ | Average acc ↑ |
|---|---|---|---|---|---|---|---|---|---|---|
| Transformer-180M | 16.89 | 45.0 | **32.5** | 39.0 | **67.1** | 59.8 | 27.9 | 51.2 | 21.8 | 42.8 |
| GDN-180M | 16.52 | **40.8** | 31.3 | 40.2 | 66.3 | **62.3** | **28.2** | 51.7 | 22.0 | 43.2 |
| Mamba-2-180M | 16.76 | 41.8 | 30.9 | 40.1 | 66.8 | 60.1 | 27.3 | **52.0** | **23.2** | 42.9 |
| **Mamba-3-SISO-180M** | **16.59** | 37.7 | **32.5** | **40.8** | 66.1 | 61.5 | 27.9 | **52.0** | 22.8 | **43.4** |
| **Mamba-3-MIMO-180M** | 16.46 | 32.1 | 34.0 | 41.0 | 66.7 | 60.6 | 27.7 | 52.9 | 22.0 | 43.5 |
| Transformer-440M | 13.03 | 21.2 | 41.7 | 50.5 | 69.9 | 67.6 | 34.6 | **56.7** | 26.0 | 49.6 |
| GDN-440M | 13.01 | **18.0** | 41.9 | 50.9 | 70.0 | 67.0 | 34.6 | 56.1 | **27.6** | 49.7 |
| Mamba-2-440M | 13.00 | 19.6 | 40.8 | **51.7** | 70.6 | 68.8 | **35.0** | 54.1 | 26.0 | 49.6 |
| **Mamba-3-SISO-440M** | **12.87** | 19.6 | 40.2 | **51.7** | **71.9** | 68.9 | 34.4 | 55.8 | 26.0 | **49.8** |
| **Mamba-3-MIMO-440M** | 12.72 | 17.1 | 43.4 | 52.8 | 70.8 | 69.6 | 35.6 | 56.3 | 28.4 | 51.0 |
| Transformer-880M | 11.42 | 15.0 | 44.7 | 57.2 | 72.6 | 71.6 | 39.2 | 57.7 | 26.8 | 52.8 |
| GDN-880M | 11.37 | **12.9** | **47.6** | 57.3 | 73.3 | 71.4 | 38.7 | **58.8** | 28.6 | 53.7 |
| Mamba-2-880M | 11.35 | 13.8 | 45.0 | 58.1 | 72.5 | 72.3 | 38.7 | 56.8 | **30.2** | 53.4 |
| **Mamba-3-SISO-880M** | **11.23** | **12.9** | 47.2 | 58.8 | 73.6 | 72.7 | 40.2 | 58.4 | 30.0 | 54.4 |
| **Mamba-3-MIMO-880M** | 11.11 | 11.8 | 49.5 | 59.2 | 73.7 | 74.7 | 41.2 | 59.9 | 28.6 | 55.3 |
| Transformer-1.5B | 10.51 | 11.1 | **50.3** | 60.6 | 73.8 | 74.0 | 40.4 | 58.7 | 29.6 | 55.4 |
| GDN-1.5B | 10.45 | **10.9** | 49.2 | 61.3 | **74.3** | 75.3 | 41.2 | 58.0 | 31.6 | 55.8 |
| Mamba-2-1.5B | 10.47 | 12.0 | 47.8 | 61.4 | 73.6 | 75.3 | 41.8 | 57.5 | **32.6** | 55.7 |
| **Mamba-3-SISO-1.5B** | **10.35** | **10.9** | 49.4 | 61.9 | 73.6 | **75.9** | 42.7 | 59.4 | 32.0 | 56.4 |
| **Mamba-3-MIMO-1.5B** | 10.24 | 10.2 | 51.7 | 62.3 | 75.3 | 76.5 | 44.5 | 60.6 | 32.6 | 57.6 |

Across all four model scales, Mamba-3 outperforms popular baselines at various downstream tasks (Table 1). We highlight that Mamba-3 does not utilize the external short convolution that has been empirically identified as an important component in many performant linear models (Gu & Dao, 2024; Yang et al., 2025c).

### 4.1.1 MIMO

We further verify the gain from MIMO by investigating its language-modeling abilities by training MIMO models with rank $R=4$ following the same settings as Section 4.1. To ensure that the total parameter count is comparable to SISO-based models, we decrease the inner dimension of the MLPs to compensate for the increase due to the MIMO projections. See Section D for details.

On both validation perplexity and our suite of language evaluation tasks (Table 1), we see significant gain when moving from SISO to MIMO for our Mamba-3 models. Namely, we achieve a significant perplexity gain of 0.11 on the 1.5B models, and Figure 2 illustrates the downward shift in our validation loss. On the language evaluation front, we see gains on most tasks when compared to SISO, resulting in an average gain of 1.2 percentage points over SISO.

### 4.1.2 RETRIEVAL CAPABILITIES

Beyond standard language modeling, an important measure for linear models is their retrieval ability—how well they can recall information from earlier in the sequence (Arora et al., 2025a;b). Unlike attention models, which can freely revisit past context with the growing KV cache, linear models must compress context into a fixed-size state. This trade-off is reflected in the Transformer baseline's substantially stronger retrieval scores. To evaluate Mamba-3 under this lens, Table 2 compares it against baselines on both real-world and synthetic needle-in-a-haystack (NIAH) tasks (Hsieh et al., 2024), using our pretrained 1.5B models from Section 4.1. We restrict the task sequence length to 2K tokens to match the training setup and adopt the cloze-style format for our real-world tasks to mirror the next-token-prediction objective, following Arora et al. (2025b; 2024).

Mamba-3 is competitive on real-world associative recall and question-answering (TQA, SQUAD) but struggles on information extraction from semi-structured or unstructured data (SWDE, FDA). On synthetic NIAH, however, Mamba-3 surpasses or matches baselines on most cases and demonstrates markedly better out-of-distribution retrieval abilities than its Mamba-2 predecessor.

**Improving Retrieval with Hybrid Models.** Because of the natural retrieval-based weaknesses of fixed state-size, we predict that linear layers will be predominately used *in hybrid models* that mitigate this downside with quadratic self-attention layers. We evaluate how Mamba-3 performs within this architectural paradigm by

Table 2: Retrieval capabilities measured by a mixture of real-world and synthetic retrieval tasks. Real-world retrieval tasks utilize cloze variants of the original datasets and are truncated to 2K length. Mamba-3 demonstrates strong associative recall, question-answering, and length generalization on needle-in-a-haystack (NIAH), but suffers with information extraction of semi-structured and unstructured data. The Transformer baseline uses RoPE which may explain its length generalization issues, and hybrid models utilize NoPE (no positional embeddings). We find a pre-gate, grouped RMSNorm can be added to Mamba-3 SISO hybrid models to improve the length generalization of the NIAH tasks at a slight decrease in real-world retrieval performance.

| Model (1.5B) | | SWDE | SQD. | FDA | TQA | NQ | Drop | NIAH-Single-1 | | | NIAH-Single-2 | | | NIAH-Single-3 | | |
|---|---|---|---|---|---|---|---|---|---|---|---|---|---|---|---|---|
| Context Length | | | | 2048 | | | | 1024 | 2048 | 4096 | 1024 | 2048 | 4096 | 1024 | 2048 | 4096 |
| Pure | Transformer | 48.9 | 46.6 | 58.4 | 67.5 | 31.7 | 26.4 | 100.0 | 100.0 | 0.0 | 92.2 | 100.0 | 0.0 | 98.6 | 99.4 | 0.0 |
| | GDN | **32.7** | 40.0 | **28.3** | 63.5 | 25.7 | 24.5 | **100.0** | **100.0** | **99.8** | **100.0** | 93.8 | 49.8 | 83.8 | 68.4 | **34.2** |
| | Mamba-2 | 30.7 | 39.1 | 23.7 | 64.3 | 25.1 | **28.5** | **100.0** | 99.6 | 62.0 | **100.0** | 53.8 | 11.8 | 95.8 | 87.4 | 13.4 |
| | **Mamba-3 SISO** | 28.5 | **40.1** | 23.4 | **64.5** | 26.5 | 27.4 | **100.0** | **100.0** | 88.2 | **100.0** | 95.4 | 50.6 | 92.4 | 81.4 | **34.2** |
| | **Mamba-3 MIMO** | 36.3 | 41.7 | 29.3 | 64.5 | 26.2 | 26.3 | 100.0 | 100.0 | 93.0 | 100.0 | 86.0 | 40.4 | 95.8 | 84.4 | 25.6 |
| Hybrid | GDN | 54.6 | **48.4** | 58.8 | 64.9 | 32.7 | **30.0** | 100.0 | 100.0 | 71.4 | 99.6 | 100.0 | 60.2 | 70.0 | 96.2 | 24.0 |
| | Mamba-2 | 58.2 | 45.6 | **71.0** | 66.1 | 33.4 | 28.1 | 100.0 | 100.0 | 3.2 | 99.6 | 98.8 | 0.0 | 98.2 | 98.0 | 0.0 |
| | Mamba-3 SISO | 58.5 | 47.0 | 65.9 | 64.8 | 33.4 | 27.0 | 100.0 | 100.0 | 36.2 | 100.0 | 100.0 | 9.4 | 99.8 | 100.0 | 8.8 |
| | Mamba-3 SISO Norm* | **58.6** | 47.3 | 52.4 | 65.7 | 33.3 | 28.5 | 100.0 | 100.0 | 100.0 | 100.0 | 100.0 | 96.0 | 99.8 | 97.2 | **56.8** |

Table 3: **Left**: Ablations on core modeling components of Mamba-3 SISO. **Right**: Formal language evaluation (scaled accuracy, %). Higher is better. SISO models are trained on short sequences and evaluated on longer lengths to test length generalization. For GDN we report the variant with eigenvalue range $[-1,1]$.

| Model Variant | ppl $\downarrow$ |
|---|---|
| Mamba-3 $-$ bias $-$ trap | 16.68 |
| Mamba-3 $-$ bias | 16.49 |
| Mamba-3 | **15.72** |
| Mamba-3 $+$ conv | 15.85 |

(a) Component ablation at 440M scale. Combining BC bias and exponential-trapezoidal discretization makes the ubiquitous short convolution optional.

| Model | Parity $\uparrow$ | Arith. w/o brackets $\uparrow$ | Arith. w/ brackets $\uparrow$ |
|---|---|---|---|
| Mamba-3 | 100.00 | 98.51 | 87.75 |
| Mamba-3 (w/ Std. RoPE) | 1.56 | 20.70 | 2.62 |
| Mamba-3 (w/o RoPE) | 2.27 | 1.49 | 0.72 |
| Mamba-2 | 0.90 | 47.81 | 0.88 |
| GDN [-1,1] | 100.00 | 99.25 | 93.50 |

(b) Performance comparison on formal language tasks. Unlike Mamba-2, Mamba-3 features state-tracking ability stemming from data-dependent RoPE embeddings.

training hybrid models with an interleaving fashion of a 5:1 ratio of linear layer to NoPE self-attention (Yang et al., 2025b). As seen in prior work (Waleffe et al., 2024), hybrid models outperform the Transformer baseline. We find that the reintroduction of the pre-output RMSNorm (pre-gate, grouped RMSNorm in Table 2) to the Mamba-3 layer improves the length generalization retrieval abilities at the slight cost of in-context, real-world retrieval tasks and is highly competitive as a linear sequence mixing backbone when mixed with self-attention. However, the ideal norm type (grouped vs default) and its placement (pre- vs post-gate) is still unclear due to competing tradeoffs (Table 8), as we find that hybrid models and their characteristics and dynamics are complex and oftentimes unintuitive, a point echoed in Cabannes et al. (2025).

## 4.2 SSM METHODOLOGY ABLATIONS

Table 3a ablates the changes that Mamba-3 introduces to core SSM components, mainly the introduction of BC bias and exponential-trapezoidal discretization. We report the pretraining test perplexity on models at the 440M scale, trained for Chinchilla optimal tokens. We find that the bias and exponential-trapezoidal SSM synergize well and make the short convolution utilized by many current linear models redundant.

We empirically demonstrate that data-dependent RoPE in Mamba-3 enables state tracking. Following Grazzi et al. (2025), we evaluate on tasks from the Chomsky hierarchy—Parity, Modular Arithmetic (without brackets), and Modular Arithmetic (with brackets)—and report scaled accuracies in Table 3b. Mamba-3 solves Parity and Modular Arithmetic (without brackets), and nearly closes the accuracy gap on Modular Arithmetic (with brackets). In contrast, Mamba-3 without RoPE, Mamba-3 with standard RoPE (Su et al., 2023), and Mamba-2 fail to learn these tasks. We use the state-tracking-enabled *GDN* variant of and observe that *Mamba-3* is competitive—matching parity and approaching its performance on both modular-arithmetic tasks. Experimental settings are covered in Section E.

## 4.3 INFERENCE EFFICIENCY TO PERFORMANCE TRADEOFF

As $d_{\text{state}}$ impacts the decode runtime for the subquadratic models considered in this paper (Section 3.3), we opt to use it as a proxy for inference speed. By plotting the validation perplexity (itself a proxy for model per-

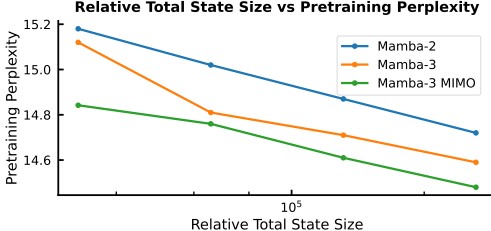

Figure 2: Exploration of state size (inference speed proxy) versus pretraining perplexity (performance proxy) across different Mamba variants. Mamba-3 improves the Pareto frontier compared to previous recurrent SISO models, while incorporating MIMO further shifts the frontier through better modeling performance without increasing state size.

| Model | FP32 | | BF16 | |
|---|---|---|---|---|
| | $d_{\text{state}}=64$ | $d_{\text{state}}=128$ | $d_{\text{state}}=64$ | $d_{\text{state}}=128$ |
| Mamba-2 | 0.295 | 0.409 | 0.127 | 0.203 |
| GDN | 0.344 | 0.423 | 0.176 | 0.257 |
| Mamba-3 (SISO) | 0.310 | 0.399 | 0.110 | 0.156 |
| Mamba-3 (MIMO) | 0.333 | 0.431 | 0.137 | 0.179 |

Table 4: Kernel latency (in milliseconds) comparison across models, precision, and $d_{\text{state}}$ values. Mamba-3 introduces minimal overhead compared to Mamba-2 and features highly efficient practical implementations. Our Mamba-3 SISO kernels are faster than reference Mamba-2 and GDN kernels at the commonly used bf16, $d_{\text{state}} = 128$ setting. Mamba-3 MIMO ($R=4$) incurs little additional cost compared to SISO.

formance) as a function of $d_{\text{state}}$, we aim to formulate a holistic picture about how the subquadratic models can trade off performance with inference speed. Figure 2 shows such a Pareto frontier for the Mamba models considered in this paper. We train a 440M parameter model to $2\times$ Chinchilla optimal tokens on the Fineweb-Edu dataset, where the model is configured with a $d_{\text{state}}$ of $\{16,32,64,128\}$. As expected, we observe an inverse correlation between validation loss and $d_{\text{state}}$. Moreover, there is a general downward shift on the Pareto frontier moving from Mamba-2 to Mamba-3, indicating a stronger model: in this setting, Mamba-3 with $2\times$ smaller state size achieves better pretraining perplexity than its Mamba-2 counterpart, resulting in a faster model with the same quality or a better model for the same speed. A further downward shift is observed when moving from the SISO variant of Mamba-3 to the MIMO variant of Mamba-3 (where we set the MIMO rank $R=4$ and decrease the MLP inner dimension to parameter match the SISO variants). We expand the comparison to include the GDN baseline in Section F, Figure 6, which also shows Mamba-3 comparing favorably to GDN.

## 4.4 Fast Mamba-3 Kernels

We complement Mamba-3's methodological advances with optimized kernels that deliver fast inference in practical settings. Specifically, we implement a new series of inference kernels for Mamba-3—using Triton for the forward (prefill) path and CuTe-DSL for decode—and compare their per-token decode latency against the released Triton kernels for Mamba-2 and GDN[2] in Table 4. The evaluation uses the setting: a decode step at batch size 128 on a single H100 for 1.5B-parameter models with model dimension 2048, state dimension $\in \{64,128\}$ in both FP32 and BF16 datatypes. Across all configurations, SISO achieves the lowest latency amongst baselines. MIMO, with its higher arithmetic intensity, increases the decoding FLOPs without significantly increasing decode runtime This indicates that our CuTe-DSL decode implementation is competitive and that the additional components of Mamba-3 (exponential-trapezoidal update, complex-valued state, and MIMO projections) are lightweight. This supports our overall inference-first perspective: the Mamba-3 admits **simple, low-latency implementation** while providing strong empirical performance.

Table 12 benchmarks end-to-end latency across different decoding sequence lengths, and additionally benchmarks prefill time for the same sequence length. The decode time is consistent with Table 4, where Mamba-3 (SISO) is fastest, Mamba-3 (MIMO) is on par with Mamba-2, and all linear methods are faster than optimized attention as sequence length grows. We also see that MIMO incurs a moderate overhead for prefill, as discussed in Section 3.3. Details of the benchmark are provided in Section H.

## 5 Conclusion And Future Work

We introduce Mamba-3, an SSM model with three axes of improvement rooted in SSM principles: (i) *improved quality*, via exponential-trapezoidal discretization; (ii) *new capabilities*, through complex SSMs that recover state tracking; and (iii) *higher inference efficiency*, with a MIMO formulation that raises arithmetic intensity. Mamba-3 delivers strong language modeling results and establishes a new Pareto frontier on the performance-efficiency axes with respect to strong baseline models. We see *hybrid Mamba-3 architectures* that integrate retrieval mechanisms as a promising path, alongside broader application of our design principles to linear-time sequence models.

---

[2]Details on each kernel DSL and the exact kernel fusion structure is provided in Appendix H.

**Acknowledgments.**

We gratefully acknowledge the support of the Schmidt Sciences AI2050 fellowship, the Google ML and Systems Junior Faculty Awards, the Google Research Scholar program, Together AI, and Cartesia AI. KL is supported by the NSF GRFP under Grant DGE2140739. We also thank Sukjun Hwang and Gaurav Ghosal for helpful feedback and discussions.

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

**LLM Usage.** We utilized Large Language Models to polish the writing in our submission as well as generate latex code for formatting tables and figures.

## A  RELATED WORK

### A.1  LINEAR-TIME SEQUENCE MIXERS

A growing body of work seeks to replace the quadratic softmax-based attention mechanism (Vaswani et al., 2017; Bahdanau et al., 2014) with linear runtime alternatives. Prominent approaches can be categorized under three broad frameworks: linear attention, test-time training, and state space models. Many nascent linear attention models aimed to approximate softmax attention through kernel feature maps (Katharopoulos et al., 2020; Choromanski et al., 2022), while recent models have discarded the feature maps for raw dot-products between queries and keys, modulated by decays or masks (Sun et al., 2023; Yang et al., 2024). More recently, fast-weight programmers Schlag et al. (2021) that modulate the state memory with key-value pairs have also fallen under the umbrella term "linear attention." Yang et al. (2025c;d) originated from this line of work and enhanced traditional linear attention by replacing the additive memory update with a delta-rule recurrence. This has further spurred on a host of work improving the efficiency and capabilities of linear models built on the delta rule (Kimi Team et al., 2025; Hu et al., 2025).

A distinct line of test-time training (TTT) work views sequence modeling as online learning task during inference. Here, the recurrent "state" is implemented as a set of weights (e.g., a linear layer or MLP) that are updated across time-steps to minimize a self-supervised inner-loop loss on the observed sequence (Sun et al., 2025). The entire TTT layer can then be viewed as a meta-learning, or bi-level optimization, scheme where the inner loop updates these fast weights while the outer loop trains the larger, encompassing network. Recent work have improved such layer's hardware efficiency (Zhang et al., 2025) and training regime (Tandon et al., 2025).

State space models (SSMs) provide linear-time sequence mixing through explicit dynamical states and efficient scan or convolution implementations, offering significant computational advantages over quadratic-time attention mechanisms. Classical linear-time invariant (LTI) SSM layers utilized structured state transition matrices, e.g., diagonal or low-rank plus diagonal, to facilitate efficient computation and stable learning of long-context tasks (Gu et al., 2022a; Smith et al., 2023; Gupta et al., 2022). The introduction of time-varying, input-dependent selectivity to SSMs in Mamba-1 (Gu & Dao, 2024) reduced the disparity between self-attention and linear models on information-dense modalities, notably language modeling. The model's success, coupled with its simplicity in using real values, led to the decline of complex-valued SSMs, which were common in LTI settings. Subsequently, Mamba-2 (Dao & Gu, 2024) formalized the connection between SSMs and (linear) attention through the structured state space duality (SSD).

### A.2  STATE TRACKING IN COMPLEX STATE SPACE MODELS

**Expressivity and State Tracking.**  Recent work characterizes the types of state that recurrent, constant-memory mixers can maintain, revealing algorithmic deficiencies in previous SSM-based models. Merrill et al. (2025) show that under finite precision, practical SSMs collapse to $TC^0$, leading to failures on tasks like permutation composition over $S_5$ unless the primitive is extended. Similarly, Yu & Erichson (2025) prove that a single-layer Mamba is not a universal approximator. Several modifications have been proposed to improve expressivity. For instance, the same work shows that a block-biased variant regains the universal approximation property with only minor changes, either through block decomposition or a channel-specific bias. Allowing negative eigenvalues or non-triangular transitions enables linear RNNs—including diagonal and Householder/DeltaNet forms—to capture parity and, under mild assumptions, regular languages (Grazzi et al., 2025). Complex-valued parameterizations provide another avenue for enhanced expressivity.

### A.3  MULTI-INPUT, MULTI-OUTPUT

S4 (Gu et al., 2022a) is a single-input, single-output LTI system where each dimension of the input was assigned its own independent SSM. S5 (Smith et al., 2023) replaced the set of SISO SSMs with a multi-input, multi-output SSM which applied directly on the entire vectorized input. This change reduced the effective state capacity but enabled the use of efficient parallel scans, forgoing the convolutional and frequency-based approaches in S4, improving pretraining speeds. While this trade-off between state capacity and modeling performance was less pronounced in LTI models, Mamba-1 (S6) (Gu & Dao, 2024) and Mamba-2 (Dao & Gu, 2024) returned to the SISO system due to the importance of a large state size expansion in the time-varying setting. The computational bottleneck associated with the increased state size was addressed with a parallel-scan, hardware-aware algorithm for Mamba-1 and a matrix multiplication-based algorithm for Mamba-2. Unlike S5, Mamba-3's MIMO structure is motivated by an inference-first perspective: to improve arithmetic intensity during the memory-bound decoding process. Accordingly, its state expansion was kept at the Mamba-1/-2 levels to maintain modeling capabilities at the cost of additional training compute.

Table 5: Table of canonical linear-time invariant discretizations (top) and custom linear-time varying discretizations derived from our exponential-adjusted framework (bottom), along with their appearance in structured SSMs used in deep learning. Our framework formalizes the prior Mamba discretization as exponential-Euler and extends it with the more expressive exponential-trapezoidal method. Discretization methods convert the continuous SSM $\dot{\boldsymbol{h}}(t) = \boldsymbol{A}(t)\boldsymbol{h}(t) + \boldsymbol{B}(t)x(t)$ into the discrete recurrence $\boldsymbol{h}_t = \alpha_t \boldsymbol{h}_{t-1} + \beta_t \boldsymbol{B}_{t-1} x_{t-1} + \gamma_t \boldsymbol{B}_t x_t$.

| Discretization Method | $\alpha_t$ | $\beta_t$ | $\gamma_t$ | Appearance |
|---|---|---|---|---|
| **Forward Euler** | $I + \Delta A$ | — | $\Delta$ | — |
| **Backward Euler** | $(I - \Delta A)^{-1}$ | — | $(I - \Delta A)^{-1}\Delta$ | — |
| **Trapezoidal** | $(I - \frac{\Delta}{2}A)^{-1}(I + \frac{\Delta}{2}A)$ | — | $(I - \frac{\Delta}{2}A)^{-1}\Delta$ | S4 |
| **Zero-Order Hold** | $\exp(\Delta A)$ | — | $A^{-1}(\exp(\Delta A) - I)$ | S4D, S5 |
| **Zero-Order Hold** | $\exp(\Delta_t A_t)$ | — | $A_t^{-1}(\exp(\Delta_t A_t) - I)$ | |
| **Exponential-Euler** | $\exp(\Delta_t A_t)$ | — | $\Delta_t$ | Mamba-1, -2[3] |
| **Exponential-Trapezoidal** | $\exp(\Delta_t A_t)$ | $(1 - \lambda_t)\Delta_t \exp(\Delta_t A_t)$ | $\lambda_t \Delta_t$ | Mamba-3 |

Moreover, the generalization from outer-product-based state update to matrix-product-based recovers a key expressive feature of SSMs in classical literature, and while there has been explorations of replacing SISO with MIMO SSMs (Smith et al., 2023), they have been limited to the LTI setting and were primarily inspired by training, not inference, efficiency. Section A discusses the history and motivations of SISO and MIMO SSM systems in greater detail.

## B  EXPONENTIAL-TRAPEZOIDAL DISCRETIZATION

**Proposition 5** (Variation of Constants (Tenenbaum & Pollard, 1985)). *Consider the linear SSM*
$$\dot{\boldsymbol{h}}(t) = A(t)\boldsymbol{h}(t) + \boldsymbol{B}(t)x(t),$$
*where $\boldsymbol{h}(t) \in \mathbb{R}^N$, $A(t) \in \mathbb{R}$ is a scalar decay, and $\boldsymbol{B}(t)x(t) \in \mathbb{R}^N$. For $\Delta_t$ discretized time grid $\tau_t = \tau_{t-1} + \Delta_t$, the hidden state satisfies Equation (9), which can then be approximated to Equation (10) with $O(\Delta_t^2)$ error. The approximation of the remaining integral on the system input can have varying error bounds depending on the method used: an example can be found in Section B.3.*

$$\boldsymbol{h}(\tau_t) = \exp\left(\int_{\tau_{t-1}}^{\tau_t} A(s)ds\right)\boldsymbol{h}(\tau_{t-1}) + \int_{\tau_{t-1}}^{\tau_t} \exp\left(\int_{\tau}^{\tau_t} A(s)ds\right)\boldsymbol{B}(\tau)x(\tau)d\tau, \tag{9}$$

$$\boldsymbol{h}_t \approx e^{\Delta_t A_t}\boldsymbol{h}_{t-1} + \int_{\tau_{t-1}}^{\tau_t} e^{(\tau_t - \tau)A_t}\boldsymbol{B}(\tau)x(\tau)d\tau. \tag{10}$$

*Proof.* Starting from the initial linear SSM, an integrating factor $z(t) := e^{\int_0^t -A(s)ds}$ is applied to facilitate integration.
$$z(t)\dot{\boldsymbol{h}}(t) = z(t)A(t)\boldsymbol{h}(t) + z(t)\boldsymbol{B}(t)x(t)$$
Keeping in mind $z'(t) = -A(t)z(t)$; through rearranging the terms and integrating between the time grid $[\tau_{t-1}, \tau_t]$
$$\int_{\tau_{t-1}}^{\tau_t} \frac{d}{d\tau}(z(\tau)\boldsymbol{h}(\tau))d\tau = \int_{\tau_{t-1}}^{\tau_t} z(\tau)\boldsymbol{B}(\tau)x(\tau)d\tau$$
results in
$$z(\tau_t)\boldsymbol{h}(\tau_t) - z(\tau_{t-1})\boldsymbol{h}(\tau_{t-1}) = \int_{\tau_{t-1}}^{\tau_t} z(\tau)\boldsymbol{B}(\tau)x(\tau)d\tau,$$
which can be arranged in a more familiar form
$$\boldsymbol{h}(\tau_t) = z(\tau_t)^{-1}z(\tau_{t-1})\boldsymbol{h}(\tau_{t-1}) + \int_{\tau_{t-1}}^{\tau_t} z(\tau_t)^{-1}z(\tau)\boldsymbol{B}(\tau)x(\tau)d\tau.$$
Substituting the integrating factor $z(\tau)$ corresponds to
$$\boldsymbol{h}(\tau_t) = \exp\left(\int_{\tau_{t-1}}^{\tau_t} A(s)ds\right)\boldsymbol{h}(\tau_{t-1}) + \int_{\tau_{t-1}}^{\tau_t} \exp\left(\int_{\tau}^{\tau_t} A(s)ds\right)\boldsymbol{B}(\tau)x(\tau)d\tau.$$

We sample the continuous underlying signal with a right-hold assumption where $\forall s \in [\tau_{t-1}, \tau_t], A(s) = A(\tau_t)$ which we refer to as $A_t$,

$$\boldsymbol{h}_t \approx \underbrace{\exp(\Delta_t A_t)\boldsymbol{h}_{t-1}}_{\text{solved analytically}} + \underbrace{\int_{\tau_{t-1}}^{\tau_t} \exp((\tau_t - \tau)A_t)\boldsymbol{B}(\tau)x(\tau)d\tau}_{\text{to be approximated}}.$$

incurring a local truncation error of order $O(\Delta_t^2)$. Thus, we have solved the exponential dynamics of the underlying ODE analytically and leave the system input integral to be approximated with any host of methods.

The same result can be shown through the Variation of Constants method.

Since $A(t)$ is scalar, the homogeneous system $\dot{\boldsymbol{h}}(t) = A(t)\boldsymbol{h}(t)$ has solution

$$\boldsymbol{h}(t) = \phi(t,s)\boldsymbol{h}(s), \qquad \phi(t,s) = \exp\left(\int_s^t A(\xi)d\xi\right).$$

The Variation of Constants formula gives us,

$$\boldsymbol{h}(t) = \phi(t,s)\boldsymbol{h}(s) + \int_s^t \phi(t,\tau)\boldsymbol{B}(\tau)x(\tau)d\tau.$$

Setting $(s,t) = (t_{k-1}, t_k)$ yields the exact $\boldsymbol{h}_t$ given $\boldsymbol{h}_{t-1}$. We approximate $\int_s^t A(\xi)d\xi$ by setting $A(\tau) \approx A_k$ over $[t_{k-1}, t_k]$, which gives us,

$$\phi(t_k, t_{k-1}) = \exp\left(\int_s^t A(\xi)d\xi\right) \approx \exp\left(\int_s^t A_k d\xi\right) = e^{\Delta_k A_k},$$

Substituting these approximations in the Variation of Constants integral, we get the approximation

$$\boldsymbol{h}_t \approx e^{\Delta_t A_t}\boldsymbol{h}_{t-1} + \int_{\tau_{t-1}}^{\tau_t} e^{(\tau_t - \tau)A_t}\boldsymbol{B}(\tau)x(\tau)d\tau.$$

$\square$

### B.1 RECOVERING PAST MAMBA LTV DISCRETIZATIONS

**ZOH.** The classical zero-order hold discretization method (Table 5) can be derived by holding $A_t, \boldsymbol{B}(\tau), x(\tau)$ as constants over the interval $[\tau_{t-1}, \tau_t]$ where the values are fixed to the right endpoint $\tau_t$.

**Exponential-Euler (Mamba-1/-2).** We can recover Mamba-1/2 discretization by approximating the state input integral with *Euler's rule* (Süli & Mayers, 2003) and holding the (right) endpoint constant throughout the interval (Fig. 1)

$$\begin{aligned}
\boldsymbol{h}_t &\approx e^{\Delta_t A_t}\boldsymbol{h}_{t-1} + (\tau_t - \tau_{t-1})e^{(\tau_t - \tau_t)A_t}\boldsymbol{B}_t x_t \\
&= e^{\Delta_t A_t}\boldsymbol{h}_{t-1} + \Delta_t \boldsymbol{B}_t x_t.
\end{aligned} \tag{11}$$

We call equation (11) the *exponential-Euler* discretization method, stemming from the exponential integration followed by Euler approximation.

### B.2 EXPONENTIAL-TRAPEZOIDAL DISCRETIZATION'S MASK MATRIX

$$\boldsymbol{L} = \begin{bmatrix} \gamma_0 & & & \\ (\gamma_0\alpha_1 + \beta_1) & \gamma_1 & & \\ \alpha_2(\gamma_0\alpha_1 + \beta_1) & (\gamma_1\alpha_2 + \beta_2) & \gamma_2 & \\ \vdots & & & \ddots \\ \alpha_{T\cdots2}(\gamma_0\alpha_1 + \beta_1) & & \cdots & \gamma_T \end{bmatrix} \tag{12}$$

$$= \begin{bmatrix} 1 & & & \\ \alpha_1 & 1 & & \\ \alpha_2\alpha_1 & \alpha_2 & 1 & \\ \vdots & & \ddots & \\ \alpha_{T\cdots1} & & \cdots & 1 \end{bmatrix} \begin{bmatrix} \gamma_0 & & & \\ \beta_1 & \gamma_1 & & \\ 0 & \beta_2 & \gamma_2 & \\ \vdots & & \ddots & \\ 0 & & \cdots & \gamma_T \end{bmatrix}. \tag{13}$$

*Proof.* When viewing the tensor contraction form, let us call $C = (T,N), B = (S,N), L = (T,S), X = (S,P)$ based on the Mamba-2 paper. With this decomposition of our mask, we can view $L = \text{contract}(TZ, ZS \rightarrow TS)(L_1, L_2)$.

The original contraction can be seen as
$$\text{contract}(TN,SN,TS,SP \rightarrow TP)(C,B,L,X)$$
We can now view it as
$$\text{contract}(TN,SN,TJ,JS,SP \rightarrow TP)(C,B,L_1,L_2,X)$$
This can be broken into the following:
$$Z = \text{contract}(SN,SP \rightarrow SNP)(B,X)$$
$$Z' = \text{contract}(JS,SNP \rightarrow JNP)(L_2,Z)$$
$$H = \text{contract}(TJ,JNP \rightarrow TNP)(L_1,Z')$$
$$Y = \text{contract}(TN,TNP \rightarrow TP)(C,H)$$
We can view this step: $\text{contract}(ZS,SNP \rightarrow ZNP)(L_2,Z)$ as a convolution of size two applied on the state-input ($B,X$ outer product) prior to the decay with the traditional SSD $L = L_1$ matrix. $\qquad\square$

### B.3 EXPONENTIAL-TRAPEZOIDAL DISCRETIZATION ERROR RATE

**Standard assumptions.** We assume that: $A(t),B(t),x(t)$ are bounded and $C^2$ on each timestep, so that $g(\tau)$ has two bounded derivatives; the map $h \mapsto A(t)h + B(t)x(t)$ is Lipschitz in $h$ which is true for linear systems; $\lambda_t$ lies in a bounded interval so that the update is zero-stable.

*Proof.* Let $g(\tau) := e^{(t_k - \tau)A_k}B(\tau)x(\tau)$ denote the integrand in the second term of Proposition 5. Since $A(t),B(t),x(t)$ are $C^2$ on $[t_{k-1},t_k]$, the function $g$ has two bounded derivatives. A second-order Taylor expansion of $g$ around $t_{k-1}$ gives us,
$$\int_{t_{k-1}}^{t_k} g(\tau)d\tau = \Delta_t g(t_{k-1}) + \frac{\Delta_t^2}{2}g'(t_{k-1}) + \frac{\Delta_t^3}{6}g''(t_{k-1}) + O(\Delta_t^4).$$

Recall that the trapezoidal approximation to this integral is given by,
$$Q_\lambda = \Delta_t\Big[(1-\lambda_t)g(t_{k-1}) + \lambda_t g(t_k)\Big].$$

Expanding $g(t_k)$ using Taylor expansion: $g(t_k) = g(t_{k-1}) + \Delta_t g'(t_{k-1}) + \frac{\Delta_t^2}{2}g''(t_{k-1}) + O(\Delta_t^3)$. Substituting this into $Q_\lambda$,
$$Q_\lambda = \Delta_t\Big[(1-\lambda_t)g(t_{k-1}) + \lambda_t g(t_k)\Big]$$
$$= \Delta_t g(t_{k-1}) + \lambda_t \Delta_t^2 g'(t_{k-1}) + \lambda_t \frac{\Delta_t^3}{2}g''(t_{k-1}) + O(\Delta_t^4).$$
Hence, the error is given by:
$$\int_{t_{k-1}}^{t_k} g(\tau)d\tau - Q_\lambda = \Big(\tfrac{1}{2} - \lambda_t\Big)\Delta_t^2 g'(t_{k-1}) + \Big(\tfrac{1}{6} - \tfrac{\lambda_t}{2}\Big)\Delta_t^3 g''(t_{k-1}) + O(\Delta_t^4).$$
Under the assumption that $\lambda_t = \frac{1}{2} + c_t\Delta_t$, where $c_t = O(1)$, then $\frac{1}{2} - \lambda_t = -c_t\Delta_t = O(\Delta_t)$ and thus the $\Delta_t^2$ term is $O(\Delta_t^3)$. Therefore,
$$\int_{t_{k-1}}^{t_k} g(\tau)d\tau - Q_\lambda = O(\Delta_t^3),$$
which yields an $O(\Delta_t^3)$ local truncation error. Since the update $h_k = e^{\Delta_t A_k}h_{k-1} + Q_\lambda$ is linear and zero–stable for bounded $\lambda_t$, standard numerical ODE results imply an $O(\Delta_t^2)$ global error. $\qquad\square$

#### B.3.1 PARALLEL REPRESENTATION OF EXPONENTIAL-TRAPEZOIDAL RECURRENCE

Our new recurrence can be instantiated as a case of SSD and has a corresponding parallel form to Equation (2). Expanding the state recurrence from $h_0 = \gamma_0 B_0 x_0$ results in $h_T = \alpha_{T\dots2}(\gamma_0\alpha_1 + \beta_1)B_0 x_0 + \cdots + \gamma_T B_T x_T$, where the SSM output is $y_T = \alpha_{T\dots2}(\gamma_0\alpha_1 + \beta_1)C_T^\top B_0 x_0 + \cdots + \gamma_T C_T^\top B_T x_T$. Unrolling these rows shows that the mask induced by the trapezoidal update is no longer a fixed averaging of endpoints (as in the classical trapezoid rule), but a *data-dependent convex combination* of the two interval endpoints.

Under the SSD framework (2) with parallel form $Y = (L \odot CB^\top)X$, Mamba-3 corresponds to a mask $L$ whose structure is a 1-semiseparable matrix composed with a 2-band matrix (13).[4] This formulation leads to a

---

[4]Incidentally, this is a special case of a 2-semiseparable matrix.

hardware-efficient matmul-based calculation of the SSM output. Finally, we note that the convolutional connection of Mamba-3 can be seen through this parallel dual form, where multiplication by the 2-band matrix in equation (13) represents convolution with weights $\beta, \gamma$. In Section B.2 we use the SSD tensor contraction machinery to prove that the parallel form is equivalent to a vanilla SSM with a convolution on the state-input.

*Remark* 5. The structured mask of Mamba-3 can be viewed as generalizing Mamba-2, which instead of the 2-band matrix has a diagonal matrix with $\gamma_t$ only (13).

### B.4 EXPONENTIAL-TRAPEZOIDAL PARAMETERIZATION

Table 6: **Ablations on $\lambda_t$ parameterization in the exponential-trapezoidal update.**

| Parameterization | Form of $\lambda_t$ | ppl $\downarrow$ |
|---|---|---|
| **Default** | $\sigma(u_t)$ | **15.72** |
| Fixed $1/2$ | $\frac{1}{2}$ | 15.76 |
| No trapezoidal (Euler) | $1$ | 15.81 |

**Setting:** All runs use the Mamba-3 (SISO) 440M model trained at Chinchilla scale, with the other architectural and optimization hyperparameters being the same as in Table 1.

The default model uses a data-dependent gate $\lambda_t = \sigma(u_t)$, where $u_t$ is a learned projection of the current input token. In Table 6, we try different parameterizations for $\lambda_t$ and find that the default parameterization empirically performs the best. Hence we choose the simpler default parameterization that does *not* enforce the $O(\frac{1}{2} + \Delta_t)$.

## C COMPLEX SSM PROOFS

### C.1 PROOF OF PROPOSITION 2

**Proposition 2** (Complex-to-Real SSM Equivalence). *Consider a complex-valued SSM*

$$\dot{\boldsymbol{h}}(t) = \mathrm{Diag}\big(A(t) + i\boldsymbol{\theta}(t)\big)\boldsymbol{h}(t) + \big(\boldsymbol{B}(t) + i\hat{\boldsymbol{B}}(t)\big)x(t), \tag{5}$$

$$y(t) = \mathrm{Re}\Big(\big(\boldsymbol{C}(t) + i\hat{\boldsymbol{C}}(t)\big)^{\top}\boldsymbol{h}(t)\Big),$$

*where $\boldsymbol{h}(t) \in \mathbb{C}^{N/2}$, $\boldsymbol{\theta}(t), \boldsymbol{B}(t), \hat{\boldsymbol{B}}(t), \boldsymbol{C}(t), \hat{\boldsymbol{C}}(t) \in \mathbb{R}^{N/2}$, and $x(t), A(t) \in \mathbb{R}$. Under exponential-Euler discretization, this system is equivalent to a real-valued SSM*

$$\boldsymbol{h}_t = e^{\Delta_t A_t} \boldsymbol{R}_t \boldsymbol{h}_{t-1} + \Delta_t \boldsymbol{B}_t x_t, \tag{6}$$

$$y_t = \boldsymbol{C}_t^{\top} \boldsymbol{h}_t,$$

*with state $\boldsymbol{h}_t \in \mathbb{R}^N$, projections*

$$\boldsymbol{B}_t := \begin{bmatrix} \boldsymbol{B}_t \\ \hat{\boldsymbol{B}}_t \end{bmatrix} \in \mathbb{R}^N, \qquad \boldsymbol{C}_t := \begin{bmatrix} \boldsymbol{C}_t \\ -\hat{\boldsymbol{C}}_t \end{bmatrix} \in \mathbb{R}^N,$$

*and a transition matrix*

$$\boldsymbol{R}_t := Block\Big(\{R(\Delta_t \boldsymbol{\theta_t}[i])\}_{i=1}^{N/2}\Big) \in \mathbb{R}^{N \times N}, \qquad R(\theta) := \begin{bmatrix} \cos(\theta) & -\sin(\theta) \\ \sin(\theta) & \cos(\theta) \end{bmatrix}.$$

*Proof.* We first present the derivation for $N = 2$; the block-diagonal structure for general even $N$ follows by grouping pairs of coordinates.

Let $h_t + i\hat{h}_t$ denote the complexified hidden state, with parameters $A(t) + i\theta(t)$ and $B(t) + i\hat{B}(t)$ for the transition and input, respectively. By the variation of constants formula (Proposition 5), applying zere-order hold and Euler's rule over a step $[t_{k-1}, t_k]$ gives

$$h_k + i\hat{h}_k = e^{\Delta_t(A_t + i\theta_t)}(h_{k-1} + i\hat{h}_{k-1}) + \Delta_t(B_t + i\hat{B}_t)x_t.$$

Expanding the exponential,

$$e^{\Delta_t(A_t + i\theta_t)} = e^{\Delta_t A_t}\Big(\cos(\Delta_t\theta_t) + i\sin(\Delta_t\theta_t)\Big),$$

so in real coordinates $\boldsymbol{h}_t = \begin{bmatrix} h_t \\ \hat{h}_t \end{bmatrix} \in \mathbb{R}^2$ the recurrence becomes

$$\boldsymbol{h}_t = e^{\Delta_t A_t} \underbrace{\begin{bmatrix} \cos(\Delta_t\theta_t) & -\sin(\Delta_t\theta_t) \\ \sin(\Delta_t\theta_t) & \cos(\Delta_t\theta_t) \end{bmatrix}}_{R(\Delta_t\theta_t)} \boldsymbol{h}_{t-1} + \Delta_t \begin{bmatrix} B_t \\ \hat{B}_t \end{bmatrix} x_t.$$

Stacking across $N/2$ such pairs yields the block-diagonal transition

$$\boldsymbol{h}_t = e^{\Delta_t A_t}\mathrm{Block}\big(\{R(\Delta_t\theta_t[i])\}_{i=1}^{N/2}\big)\boldsymbol{h}_{t-1} + \Delta_t\begin{bmatrix}\boldsymbol{B}_t\\ \hat{\boldsymbol{B}}_t\end{bmatrix}x_t.$$

For the output,

$$y_t = \mathrm{Re}\Big((\boldsymbol{C}_t + i\hat{\boldsymbol{C}}_t)^\top(h_t + i\hat{h}_t)\Big) = \begin{bmatrix}\boldsymbol{C}_t\\ -\hat{\boldsymbol{C}}_t\end{bmatrix}^\top \boldsymbol{h}_t,$$

which defines the real projection $\boldsymbol{C}_t \in \mathbb{R}^N$ in the proposition. This proves the equivalence between complex SSM and the real block-diagonal system with rotations. $\qquad\square$

## C.2 Proof of Proposition 3

**Proposition 3** (Complex SSM, Data-Dependent RoPE Equivalence). *Under the notation established in Proposition 2, consider the real SSM defined in equation (6) unrolled for $T$ time-steps. The output of the above SSM is equivalent to that of a vanilla scalar transition matrix-based SSM (11) with a data-dependent rotary embedding applied on the $\boldsymbol{B},\boldsymbol{C}$ components of the SSM, as defined by:*

$$\boldsymbol{h}_t = e^{\Delta_t A_t}\boldsymbol{h}_{t-1} + \left(\prod_{i=0}^t \boldsymbol{R}_i^\top\right)\boldsymbol{B}_t x_t, \qquad y_t = \left[\left(\prod_{i=0}^t \boldsymbol{R}_i^\top\right)\boldsymbol{C}_t\right]^\top \boldsymbol{h}_t \tag{7}$$

*where the matrix product represents right matrix multiplication, e.g., $\prod_{i=0}^1 \boldsymbol{R}_i = \boldsymbol{R}_0\boldsymbol{R}_1$. We refer to the usage of a transformed real-valued SSM to compute the complex SSM as the "RoPE trick."*

*Proof.* Consider the SSM

$$\boldsymbol{h}_t = e^{\Delta_t A_t}\boldsymbol{R}_t\boldsymbol{h}_{t-1} + \boldsymbol{B}_t x_t, \qquad y_t = \boldsymbol{C}_t^\top \boldsymbol{h}_t, \tag{14}$$

where (as in Proposition 3) $A_t \in \mathbb{R}$ is a scalar (so that $e^{\Delta_t A_t}$ is a scalar and commutes with rotations), and $\boldsymbol{R}_t$ is block-diagonal orthogonal/unitary, hence $\boldsymbol{R}_t^{-1} = \boldsymbol{R}_t^\top$.

Unrolling the recurrence with the convention that an empty product is the identity,

$$\boldsymbol{h}_t = \sum_{i=0}^t \left(\prod_{s=i+1}^t e^{\Delta_s A_s}\boldsymbol{R}_s\right)\boldsymbol{B}_i x_i. \tag{15}$$

Thus

$$y_t = \boldsymbol{C}_t^\top \boldsymbol{h}_t = \sum_{i=0}^t \boldsymbol{C}_t^\top\left(\prod_{s=i+1}^t e^{\Delta_s A_s}\boldsymbol{R}_s\right)\boldsymbol{B}_i x_i. \tag{16}$$

Using its unitary property,

$$\prod_{s=i+1}^t \boldsymbol{R}_s = \left(\prod_{s=0}^t \boldsymbol{R}_s\right)\left(\prod_{s=0}^i \boldsymbol{R}_s\right)^{-1} = \left(\prod_{s=0}^t \boldsymbol{R}_s\right)\left(\prod_{s=0}^i \boldsymbol{R}_s^\top\right).$$

Since $e^{\Delta_s A_s}$ are scalars, they commute with rotations; hence

$$y_t = \sum_{i=0}^t \boldsymbol{C}_t^\top\left(\prod_{s=0}^t \boldsymbol{R}_s\right)\left(\prod_{s=i+1}^t e^{\Delta_s A_s}\right)\left(\prod_{s=0}^i \boldsymbol{R}_s^\top\right)\boldsymbol{B}_i x_i \tag{17}$$

$$= \left[\left(\prod_{s=0}^t \boldsymbol{R}_s^\top\right)\boldsymbol{C}_t\right]^\top \sum_{i=0}^t \left(\prod_{s=i+1}^t e^{\Delta_s A_s}\right)\left(\prod_{s=0}^i \boldsymbol{R}_s^\top\right)\boldsymbol{B}_i x_i. \tag{18}$$

Define the rotated parameters $\overline{\boldsymbol{C}}_t := \left(\prod_{s=0}^t \boldsymbol{R}_s^\top\right)\boldsymbol{C}_t$ and $\overline{\boldsymbol{B}}_i := \left(\prod_{s=0}^i \boldsymbol{R}_s^\top\right)\boldsymbol{B}_i$. Then

$$y_t = \overline{\boldsymbol{C}}_t^\top \sum_{i=0}^t \left(\prod_{s=i+1}^t e^{\Delta_s A_s}\right)\overline{\boldsymbol{B}}_i x_i. \tag{19}$$

Equivalently, introducing the rotated state $\tilde{\boldsymbol{h}}_t := \left(\prod_{s=0}^t \boldsymbol{R}_s^\top\right)\boldsymbol{h}_t$,

$$\tilde{\boldsymbol{h}}_t = e^{\Delta_t A_t}\tilde{\boldsymbol{h}}_{t-1} + \overline{\boldsymbol{B}}_t x_t, \qquad y_t = \overline{\boldsymbol{C}}_t^\top \tilde{\boldsymbol{h}}_t, \tag{20}$$

$\qquad\square$

### C.3 PROOF OF PROPOSITION 4

**Proposition 4** (Rotary Embedding Equivalence with Exponential-Trapezoidal Discretization). *Discretizing a complex SSM with the exponential-trapezoidal rule (Proposition 1) yields the recurrence*

$$\boldsymbol{h}_t = \alpha_t \boldsymbol{h}_{t-1} + \beta_t \left(\prod_{i=0}^{t-1} \boldsymbol{R}_i^\top\right) \boldsymbol{B}_{t-1} x_{t-1} + \gamma_t \left(\prod_{i=0}^{t} \boldsymbol{R}_i^\top\right) \boldsymbol{B}_t x_t,$$

$$y_t = \left[\left(\prod_{i=0}^{t} \boldsymbol{R}_i^\top\right) \boldsymbol{C}_t\right]^\top \boldsymbol{h}_t. \tag{8}$$

*Here $\boldsymbol{R}_t$ is the block-diagonal rotation matrix defined in Proposition 3.*

*Proof.* We begin from the complex SSM (as in Prop. 2)

$$\dot{\boldsymbol{h}}(t) = \mathrm{Diag}(A(t) + i\boldsymbol{\theta}(t))\boldsymbol{h}(t) + \left(\boldsymbol{B}(t) + i\hat{\boldsymbol{B}}(t)\right) x(t),$$

$$y(t) = \mathrm{Re}\left((\boldsymbol{C}(t) + i\hat{\boldsymbol{C}}(t))^\top \boldsymbol{h}(t)\right),$$

where $A(t) \in \mathbb{R}$ is a scalar and $\boldsymbol{\theta}(t), \boldsymbol{B}(t), \hat{\boldsymbol{B}}(t), \boldsymbol{C}(t), \hat{\boldsymbol{C}}(t) \in \mathbb{R}^{N/2}$.

Recall from Prop. 5,

$$\boldsymbol{h}_t \approx e^{\Delta_t(A_t + i\boldsymbol{\theta}_t)} \boldsymbol{h}_{t-1} + \int_{\tau_{t-1}}^{\tau_t} e^{(\tau_t - \tau)(A_t + i\boldsymbol{\theta}_t)} \left(\boldsymbol{B}(\tau) + i\hat{\boldsymbol{B}}(\tau)\right) x(\tau) d\tau.$$

Applying Prop. 1 to the above integral, we get

$$\boldsymbol{h}_t = e^{\Delta_t(A_t + i\boldsymbol{\theta}_t)} \boldsymbol{h}_{t-1} + \beta_t e^{i\Delta_t \boldsymbol{\theta}_t} \left(\boldsymbol{B}_{t-1} + i\hat{\boldsymbol{B}}_{t-1}\right) x_{t-1} + \gamma_t \left(\boldsymbol{B}_t + i\hat{\boldsymbol{B}}_t\right) x_t, \tag{21}$$

where

$$\alpha_t := e^{\Delta_t A_t}, \qquad \beta_t := (1 - \lambda_t)\Delta_t e^{\Delta_t A_t}, \qquad \gamma_t := \lambda_t \Delta_t,$$

Since $e^{\Delta_t(A_t + i\boldsymbol{\theta}_t)} = \alpha_t e^{i\Delta_t \boldsymbol{\theta}_t}$ and as shown in Prop. 2, multiplication by $e^{i\Delta_t \boldsymbol{\theta}_t}$ is a block-diagonal rotation in real coordinates, we get the real $N$-dimensional recurrence

$$\boldsymbol{h}_t = \alpha_t \boldsymbol{R}_t \boldsymbol{h}_{t-1} + \beta_t \boldsymbol{R}_t \boldsymbol{B}_{t-1} x_{t-1} + \gamma_t \boldsymbol{B}_t x_t, \tag{22}$$

$$y_t = \boldsymbol{C}_t^\top \boldsymbol{h}_t,$$

where $\boldsymbol{R}_t := \mathrm{Block}\left(\{R(\Delta_t \boldsymbol{\theta}_t[i])\}_{i=1}^{N/2}\right)$ where $R(\theta) := \begin{bmatrix} \cos\theta & -\sin\theta \\ \sin\theta & \cos\theta \end{bmatrix}$, and projections $\boldsymbol{B}_t := \begin{bmatrix} \boldsymbol{B}_t \\ \hat{\boldsymbol{B}}_t \end{bmatrix}, \boldsymbol{C}_t := \begin{bmatrix} \boldsymbol{C}_t \\ -\hat{\boldsymbol{C}}_t \end{bmatrix}$. Note that $\boldsymbol{R}_t$ is orthogonal, so $\boldsymbol{R}_t^{-1} = \boldsymbol{R}_t^\top$.

We define the following,

$$\tilde{\boldsymbol{h}}_t := \left(\prod_{s=0}^{t} \boldsymbol{R}_s^\top\right) \boldsymbol{h}_t, \qquad \overline{\boldsymbol{B}}_t := \left(\prod_{s=0}^{t} \boldsymbol{R}_s^\top\right) \boldsymbol{B}_t, \qquad \overline{\boldsymbol{C}}_t := \left(\prod_{s=0}^{t} \boldsymbol{R}_s^\top\right) \boldsymbol{C}_t.$$

Left-multiplying equation (22) by $\prod_{s=0}^{t} \boldsymbol{R}_s^\top$ and using $\boldsymbol{R}_t^\top \boldsymbol{R}_t = I$,

$$\tilde{\boldsymbol{h}}_t = \alpha_t \tilde{\boldsymbol{h}}_{t-1} + \beta_t \overline{\boldsymbol{B}}_{t-1} x_{t-1} + \gamma_t \overline{\boldsymbol{B}}_t x_t,$$

$$y_t = \overline{\boldsymbol{C}}_t^\top \tilde{\boldsymbol{h}}_t.$$

This is a vanilla scalar-transition SSM with data-dependent rotary embeddings absorbed into $\boldsymbol{B}, \boldsymbol{C}$ via cumulative products of $\boldsymbol{R}_s^\top$. $\qquad \square$

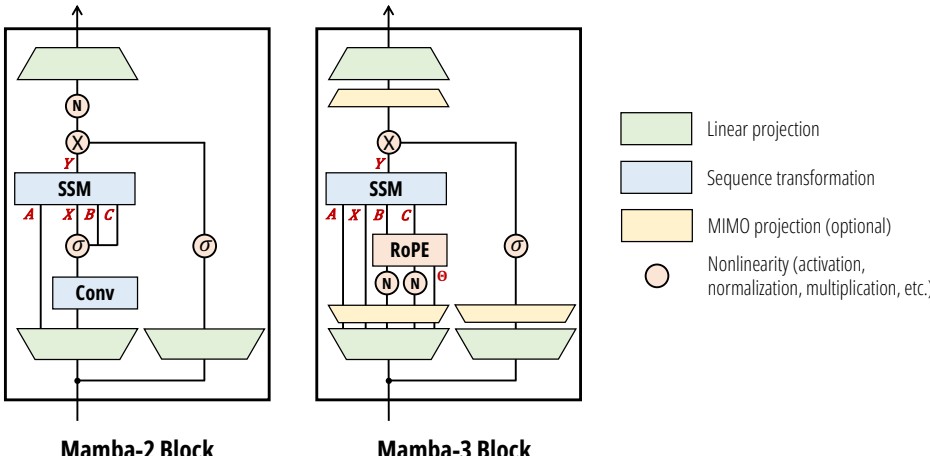

Figure 3: Contrasting Mamba-2 and Mamba-3 Architectures: Key updates include exponential-trapezoidal discretization, data-dependent RoPE embeddings, MIMO projections, QK normalization, and learnable biases.

Table 7: Arithmetic Intensity for (a) SISO, (b) MIMO. The batch and head dimensions cancel out. The arithmetic intensity of MIMO increases linearly with rank $R$, enabling better hardware utilization during memory-bound phases like decode. Here $N$ is the state size (expansion factor) and $P$ is the head dimension. For Mamba-3, typically $R \ll N, P$.

| Input | Output | FLOPs | Arithmetic Intensity | Input | Output | FLOPs | Arithmetic Intensity |
|-------|--------|-------|----------------------|-------|--------|-------|----------------------|
| $H_t : (N,P)$ $x_t : (P)$ $a_t : (1)$ $b_t : (N)$ $c_t : (N)$ | $y_t : (P)$ | $5NP - P$ | $\frac{5NP - P}{2(1 + 2N + P + NP)}$ $\approx 2.5 = \Theta(1)$ | $H_t : (N,P)$ $x_t : (P,R)$ $a_t : (1)$ $b_t : (N,R)$ $c_t : (N,R)$ | $y_t : (P,R)$ | $4NPR +$ $NP - PR$ | $\frac{4NPR + NP - PR}{2(1 + 2NR + PR + NP)}$ $\Theta(\min(N,P,R))$ $= \Theta(R), R \ll N, P$ |
| | (a) SISO (2-byte data). | | | | (b) MIMO (2-byte data). | | |

## D  MIMO FOR MAMBA-3

**Training MIMO SSMs.**    While the MIMO formulation is motivated by *inference* efficiency, the *training* algorithms for SSMs (including our developments in Section 3.1, Section 3.2) are generally developed for SISO models. We note that MIMO SSMs can be expressed in terms of $R^2$ SISO SSMs, where $R$ SISO SSMs sharing the same recurrence are summed for each of the $R$ MIMO outputs. In particular, if we define $C_t^{(i)} \in \mathbb{R}^N, B_t^{(j)} \in \mathbb{R}^N, x_t^{(j)} \in \mathbb{R}$, where $i,j \in \{0, ..., R-1\}$, then we have

$$h_t^{(j)} \leftarrow \alpha_t h_{t-1}^{(j)} + B_t^{(j)} x_t^{(j)}, \qquad h_t = \sum_{j=0}^{R-1} h_t^{(j)} \tag{23}$$

$$y_t^{(i)} \leftarrow \left( C_t^{(i)} \right)^\top h_t \tag{24}$$

Consequently, improvements to standard SISO-based SSM models can be directly applied to MIMO models as the underlying SISO training algorithms can be utilized as a black-box. The general sequence-to-sequence formulation requires calling the SISO algorithm $R^2$ times as a black box. On the other hand, in the recurrent form the computations for equation (23) and equation (24) can be performed sequentially and incurs an $R$-times overhead (recall the discussion on MIMO decoding FLOPs).

**Chunked Algorithm for MIMO SSMs.**    Many modern SISO recurrent models including Mamba-2 are computed using a parallelized *chunk-wise* algorithm, where the sequence is divided into chunks of length $C$ and a parallelizable (but asymptotically slower) algorithm is run within each chunk, while recurrence

is computed between chunks. In these cases, we can reduce the $R^2$ increase in compute to a factor of $R$ by exploiting the structure of the chunked algorithm in a more fine-grained way.

All instantiations of state space duality (SSD) for example have chunked algorithms using the quadratic dual algorithm intra-chunk. Dao & Gu (2024) showed that in the case when the state dimension and head dimension are similar, the chunk size can also be set to the same value and the overall algorithm has linear cost. More precisely, the intra-chunk computation incurs $2 \cdot \left( \frac{T}{C} C^2 N + \frac{T}{C} CPN \right)$ FLOPs and inter-chunk computation incurs $4 \cdot \frac{T}{C} NPC$ FLOPs (ignoring negligible terms). Setting $C = P = N$ yields a total FLOP count of $8TN^2$.

The chunked algorithm for SSD can be naturally generalized into MIMO SSMs. In such a case, the intra-chunk computation incurs $2 \cdot \left( \frac{T}{C} (CR)^2 N + \frac{T}{C} (CR) PN \right)$ FLOPs and inter-chunk computation incurs $4 \cdot \frac{T}{C} NP(CR)$ FLOPs. Thus, setting $CR = N = P$ yields a total FLOP count of $8TRN^2$, an $R$-fold increase in FLOP count. Intuitively, setting MIMO chunk size as $\frac{1}{R}$ times the SISO chunk size, i.e., $C_{\text{MIMO}} \leftarrow \frac{1}{R} C_{\text{SISO}}$, maintains the SISO intra-chunk FLOP count while increasing the number of chunks by a factor of $R$, resulting in an overall $R$-times increase in FLOP count.

The training speed of algorithms in practice depend on details of the kernel implementation strategy, architectural choices such as how the MIMO parameters are instantiated, and problem dimensions, but should be no more than $R$ times slower. Our released Triton Mamba-3 SISO kernels are roughly on par with the Triton Mamba-2 kernels, and MIMO kernels incur a slowdown. Table 4 benchmarks the prefill speed of various kernels which is a a proxy for the forward pass of the training kernel.

**MIMO Instantiation.** Among various choices for MIMO parameterizations, Mamba-3's approach achieves a balance that preserves the state size and number of SSMs of its SISO counterpart, while avoiding excessive growth in parameter count. The naive conversion of a SISO SSM to a rank $R$ MIMO SSM would incur a $R\times$ increase in parameters as all projections that model the inputs to the SSM, $B, C, x$, would increase. Block-level components, such as the gate $Z$ (which so far has been ignored for simplicity) and output $y$ projection would also be impacted. This influx in parameter count would be intractable at larger model scales. To counteract this, we make the following change. Mamba's multi-value attention (MVA) head structure results in shared $B, C$ across heads, so these component's projections can be directly converted to incorporate the new MIMO rank $R$ with only a slight increase in parameter count from $DN$ to $DNR$ for the entire layer. However, the SSM input $x_t$, output $y_t$, and gate $z_t$ are unique per head and therefore dominate the parameter count. Here, directly adjusting the projections would increase the parameter count from $DP$ to $DPR$ for *each head*. Instead, we keep the original SISO projection and element-wise scale each dimension of the projected output to size $R$ with a learnable, data-independent vector, resulting in $DP + PR$ parameters for each head. This mitigates the multiplicative increase to a more reasonable additive parameter count increase. Section D details the parameterization, and all MIMO-variants in our paper are parameter-matched to their SISO counterparts by reducing the MLP width.

**MIMO Formal Parameterization.** With a given batch, head, and sequence position $t$, consider the input $U_t \in \mathbb{R}^D$. Also denote $P, R \in \mathbb{N}$ as the head dimension and MIMO rank, respectively. We first obtain SSM parameters via a set of projections defined in terms of tensor contraction notation as follows:

$$\boldsymbol{B}_t = \text{contract}(DNR, D \to NR)(\boldsymbol{W}_B, \boldsymbol{U}_t) \qquad \boldsymbol{C}_t = \text{contract}(DNR, D \to NR)(\boldsymbol{W}_C, \boldsymbol{U}_t),$$
$$\boldsymbol{X}'_t = \text{contract}(PD, D \to P)(\boldsymbol{W}_{X'}, \boldsymbol{U}_t) \qquad \boldsymbol{X}_t = \text{contract}(PR, P \to PR)(\boldsymbol{W}_X, \boldsymbol{X}'_t),$$

where $\boldsymbol{W}_B, \boldsymbol{W}_C, \boldsymbol{W}_{X'}, \boldsymbol{W}_X$ are model parameters. Additionally, we obtain the residual gate term $\boldsymbol{Z}_t$ in the same manner as $\boldsymbol{X}_t$ with weights $\boldsymbol{W}_{Z'}$ and $\boldsymbol{W}_Z$ to temper the increase in the parameter count. The state update and the SSM output are then computed via the following MIMO SSM:

$$\boldsymbol{H}_t = a_t \boldsymbol{H}_{t-1} + \boldsymbol{B}_t \boldsymbol{X}_t^\top \in \mathbb{R}^{N \times P}, \qquad \boldsymbol{Y}_t = \boldsymbol{H}_t^\top \boldsymbol{C}_t \in \mathbb{R}^{P \times R}.$$

Intermediate output $\boldsymbol{Y}'_t$ is obtained by the residual function $\phi$, $\boldsymbol{Y}'_t \leftarrow \phi(\boldsymbol{Y}_t, \boldsymbol{Z}_t)$ where $\phi(\boldsymbol{Y}_t, \boldsymbol{Z}_t) \coloneqq \boldsymbol{Y}_t \circ \text{SiLU}(\boldsymbol{Z}_t)$ in our case. Finally, the layer output $\boldsymbol{O}_t \in \mathbb{R}^D$ is computed via the following down projections:

$$\boldsymbol{O}'_t = \text{contract}(PR, PR \to P)(\boldsymbol{W}_{O'}, \boldsymbol{Y}'_t) \qquad \boldsymbol{O}_t = \text{contract}(PD, P \to D)(\boldsymbol{W}_O, \boldsymbol{O}'_t).$$

This formulation enhances the existing Mamba-3 architecture by providing a lightweight parameterization that transforms the set of independent SISO SSMs within each head into a set of MIMO SSMs. Here, we note that the hardware-efficient chunking technique employed by Mamba-2 for pretraining can be applied with little change, as the decay factor $\alpha_t$ is tied across the MIMO dimension $r$.

**MIMO Parameter Matching.** The MIMO variant of Mamba3 incurs additional parameters compared to its SISO counterpart. We therefore reduce the hidden dimension of the MLP layers to parameter-match the SISO variants as follows:

| Model | 180M | 440M | 880M | 1.5B |
|---|---|---|---|---|
| SISO MLP dim | 1,500 | 2,048 | 3,072 | 4,096 |
| MIMO MLP dim ($R=4$) | 1,264 | 1,792 | 2,800 | 3,824 |

*Remark 6.* For simplicity, all discussion in this section was for simpler 2-term recurrences such as that arising from exponential-Euler discretization; the generalization to the 3-term exponential-trapezoidal recurrence is similar.

# E  EXPERIMENTAL DETAILS

**Language Modeling.** Our pretraining procedures follow that of Dao & Gu (2024)'s section D.2. All models at each scale follow the same procedure and were trained with bfloat16. The Mamba family of models were trained using the standard expand factor of 2 and a dstate of 128 and head dimension of 64. The Transformer baselines follows Dao & Gu (2024), and the GDN baselines follow (Yang et al., 2025c) where $q, k_{\dim} = 128, v_{dim} = 256$. We utilize the Llama-3.1 tokenizer (Grattafiori et al., 2024) for all models.

We utilize LM Evaluation Harness (Gao et al., 2024) to test the zero-shot languag modeling capabilities of our pretrained model on LAMBADA (OpenAI version) (Paperno et al., 2016), HellaSwag (Zellers et al., 2019), PIQA (Bisk et al., 2019), Arc-Easy/Arc-Challenge (Clark et al., 2018), WinoGrande (Sakaguchi et al., 2019), and OpenBookQA (Mihaylov et al., 2018).

**Real-World and Synthetic Retrieval.** For our real-world retrieval tasks, we evaluate on the common suite consisting of SWDE (Arora et al., 2025b), SQUAD (Rajpurkar et al., 2018), FDA (Arora et al., 2025b), TriviaQA (Joshi et al., 2017), NQ (Kwiatkowski et al., 2019), and DROP (Dua et al., 2019). We utilize the cloze-formatted version of the aforementioned tasks provided by Arora et al. (2025b; 2024), as the original datasets are in a question-answering format, making it challenge for solely pretrained models. All tasks were truncated to match the training context length. The synthetic NIAH tasks (Hsieh et al., 2024) were also run with LM Evaluation Harness.

**State-Tracking Synthetics.** Training follows a sequence length curriculum that sets the minimum length to 3 and progresses the maximum length from 40 to 160. Final models are evaluated at 256 length. Each curriculum runs for $10^4$ steps with batch size 256. We use 1 layer models for Parity and 3 layer models for Modular-arithmetic tasks. The state size is chosen to be 64, and we sweep $d_{\text{model}} \in \{32, 64\}$ and 8 learning rates logarithmically spaced between $10^{-4}$ and $10^{-2}$, reporting the best validation accuracy.

# F  ADDITIONAL EXPERIMENTAL RESULTS

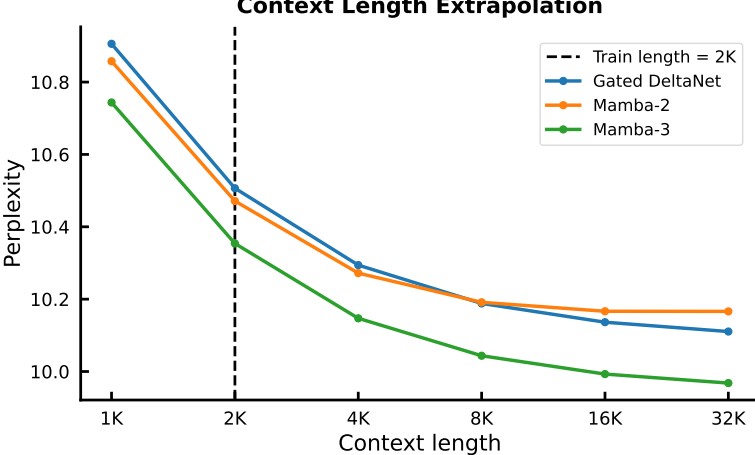

Figure 4: Pretrained 1.5B models' performance on the held-out FineWeb-Edu test set at varying context lengths. Mamba-3 exhibits strong length extrapolation while Mamba-2 falters at longer contexts.

Table 8: Ablations of optional norm type (grouped vs default) and placement (pre- vs post-gate) on pretrained hybrid Mamba-3 SISO models at the 1.5B scale. All models have BCNorm. No additional norm demonstrates the strongest in-context retrieval performance on average, while pre-gate, grouped RMS results in the best performance on synthetic retrieval, especially on lengths longer than its training context.

| Mamba-3 Norm Type | LM Avg. | SWDE | SQD. | FDA | TQA | NQ | Drop | NIAH-Single-1 | | | NIAH-Single-2 | | | NIAH-Single-3 | | |
|---|---|---|---|---|---|---|---|---|---|---|---|---|---|---|---|---|
| Context Length | — | | | 2048 | | | | 1024 | 2048 | 4096 | 1024 | 2048 | 4096 | 1024 | 2048 | 4096 |
| No Norm | 56.4 | 58.5 | 47.0 | 65.9 | 64.8 | **33.4** | 27.0 | 100.0 | 100.0 | 36.2 | 100.0 | 100.0 | 9.4 | **99.8** | **100.0** | 8.8 |
| Post-Gate Default RMS | **56.5** | 54.5 | 46.6 | 61.9 | 65.4 | 31.9 | **29.2** | 100.0 | 100.0 | 100.0 | 100.0 | 99.8 | 49.2 | 87.6 | 94.0 | 62.0 |
| Pre-Gate Default RMS | 55.9 | 55.4 | 46.9 | **67.3** | 65.4 | 33.0 | 28.1 | 100.0 | 100.0 | 86.2 | 100.0 | 100.0 | **97.8** | 99.2 | 97.8 | **90.2** |
| Post-Gate Grouped RMS | 56.2 | 51.4 | 46.7 | 56.8 | 64.2 | 30.4 | 27.6 | 100.0 | 100.0 | 79.4 | 100.0 | 100.0 | 65.8 | 93.8 | 97.0 | 9.6 |
| Pre-Gate Grouped RMS | 56.1 | **58.6** | **47.3** | 52.4 | **65.7** | 33.3 | 28.5 | 100.0 | 100.0 | 100.0 | 100.0 | 100.0 | 96.0 | **99.8** | 97.2 | 56.8 |

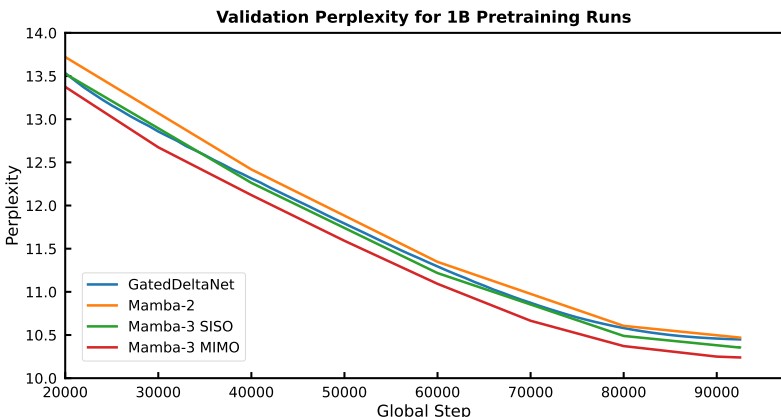

Figure 5: Mamba-3 demonstrates better pretraining performance compared to strong baselines like Mamba-2 and Gated Deltanet. These are the validation perplexity on FineWeb-Edu of our fully pretrained 1.5B models.

We also compare the effectiveness of state size usage of Mamba variants to a Gated DeltaNet baseline in Figure 6. We highlight the difficulty of directly comparing GDN versus Mamba-style models due to the differing head structure (multi-head for GDN compared to multi-value for Mamba). Our experiments hold GDN's $v_{expand}$ to 2 and decrease the head dimension accordingly to vary the relative total state size. Similar to Figure 2, we train 440M models to $2\times$ Chinchilla tokens ($40\times$ token-to-parameter ratio) and sweep across $d_{\text{state}} = \{32, 64, 128\}$ for the Mamba models and $d_{\text{head dim}} = \{32, 64, 128\}$ for GDN. We parameter match all models.

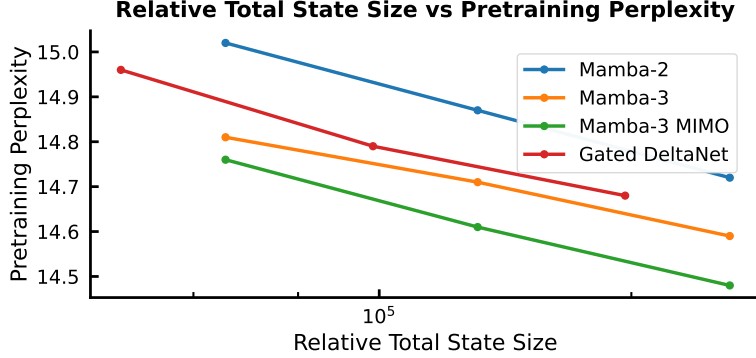

Figure 6: Exploration of state size (inference speed proxy) versus pretraining perplexity (performance proxy). Mamba-3 and Mamba-3 MIMO continue set the Pareto frontier.

## G  ARCHITECTURE ABLATIONS

We explore our model architecture's ablation in this section. All models are trained at the 440M scale to Chinchilla optimal number of tokens ($20\times$ tokens to parameters) with the same experimental procedures as our pretrained models as covered in Section E unless otherwise stated.

$B$,$C$ **Bias Parameterization.**  The Mamba-3 model's separate $B$ and $C$ biases are head-specific and channel-wise and added to both $B$ and $C$ after the QK-Norm. While the biases in the final Mamba-3 model are trainable, data-independent parameters and initialized to all ones, we explore various bias parameterizations in Table 9a. We find our models are not very sensitive to the initialization of the biases as long as they are positive. We choose the all-ones initialization due to it's simplicity.

We also explore the impact removing the $B$ or $C$ bias on performance in Table 9b (bias is initialized with our default parameterization when utilized). Unlike in Yu & Erichson (2025), which finds that $B$ bias by itself is able to improve performance on Mamba-1, our experiments find that only having $B$ bias hurts performance slightly and that $B$ and $C$ biases have synergetic properties.

| Bias Init. | Trainable | ppl $\downarrow$ |
|---|---|---|
| 1.0 | ✓ | 15.72 |
| 0.0 | ✓ | 16.57 |
| 1.0 | ✗ | 15.80 |
| $\mathcal{U}(0,1)$ | ✓ | 15.76 |
| $\mathcal{U}(-1,1)$ | ✓ | 16.07 |

(a) Effect of parameterization of the $B$ and $C$ bias on model performance, measured by pretraining perplexity. We find our default initialization of all-ones (first row) provides the best performance, but performance is not sensitive as long as biases are positive.

| $B$ Bias | $C$ Bias | ppl $\downarrow$ |
|---|---|---|
| ✗ | ✗ | 16.52 |
| ✓ | ✗ | 16.68 |
| ✗ | ✓ | 15.98 |
| ✓ | ✓ | 15.69 |

(b) Applying a bias to both $B$ and $C$ leads to the best performance. Only applying $B$ bias (Block-Biased (Yu & Erichson, 2025) Mamba-3 variant) does not provide significant gains over the no-bias baseline.

Table 9: Ablations on $B$,$C$ bias initialization (left) and presence (right) for Mamba-3.

## H  INFERENCE KERNEL LATENCY ANALYSIS

### H.1  KERNEL IMPLEMENTATIONS AND FUSION STRUCTURE

In Table 4, we detail the DSL (Triton, TileLang, CuTe, PyTorch) and the fusion level of the kernels used in our latency analysis. For Mamba-2 and Gated DeltaNet (GDN), we directly use the publicly released Triton kernels from the respective authors. For Mamba-3, we implement new inference kernels with a comparable fusion structure: the forward SISO uses a Triton kernel fused with rotary position embeddings and the forward MIMO uses a TileLang kernel with the same fusion level while the decode path uses a CuTe kernel fused with gating and MIMO projection.

In Tables 10 and 11, we abbreviate IP = input projection, Conv = 1D convolution, Gate = gating, OP = output projection. Colors indicate implementation backend (Torch, Triton, TileLang, CuTe).

Table 10: Kernel DSL and fusion structure for **forward** (prefill) kernels.

| Model (Forward) | Kernel DSL | Fusion Level |
|---|---|---|
| Mamba-2 | Triton | IP, Conv, SSM, Gate+OP |
| Gated DeltaNet | Triton | IP, Conv, Chunked Delta, Gate+OP |
| Mamba-3 (SISO) | Triton | IP, SSM+Rotary, Gate+OP |
| Mamba-3 (MIMO) | TileLang | IP, SSM+Rotary, Gate+OP |

### H.2  EXTENDED PREFILL AND PREFILL+DECODE LATENCY MEASUREMENTS

**Models.**  We benchmark Mamba-3 1.5B (SISO), Mamba-2 1.5B, Gated DeltaNet (GDN) 1.5B, and a strong Transformer baseline implemented via the vLLM engine (v0.11.0) with Llama-3.2 1B.[5] All recurrent models are trained at the 1.5B scale with $d_{model} = 2048$ and 24 layers. For Mamba variants we set state size as 128 and head dimension 64; for GDN we use QK head dimension as 128.

---
[5]https://huggingface.co/meta-llama/Llama-3.2-1B

Table 11: Kernel DSL and fusion structure for **decode** kernels.

| Model (Decode) | Kernel DSL | Fusion Level |
|---|---|---|
| Mamba-2 | Triton | IP, Conv, SSM, Gate+OP |
| Gated DeltaNet | Triton | IP, Conv, Recurrent Delta, Gate+OP |
| Mamba-3 (SISO) | CuTe + Triton | IP, Rotary, SSM+Gate, OP |
| Mamba-3 (MIMO) | CuTe + Triton | IP, Rotary, SSM+Gate+MIMO, OP |

**Setting.** Sequence lengths were swept over $L \in \{512, 1024, 2048, 4096, 16384\}$ for prefill, with an equal number of tokens decoded. For sequence lengths $\{512, 1024, 2048, 4096\}$, we use batch size of 128; for sequence lengths $\{16384\}$, we use batch size of 16. We use a single H100-SXM 80GB GPU and report wall-clock times (in seconds) over 3 repetitions.

Table 12: Prefill and Prefill+Decode latency across sequence lengths. Mamba-3 adds minimal overhead to its forward-pass and retains competitive decode latencies. Details in Section H.

| Model | 512 tokens | | 1024 tokens | | 2048 tokens | | 4096 tokens | | 16384 tokens | |
|---|---|---|---|---|---|---|---|---|---|---|
| | Prefill | Prefill+Dec | Prefill | Prefill+Dec | Prefill | Prefill+Dec | Prefill | Prefill+Dec | Prefill | Prefill+Dec |
| vLLM (Llama-3.2-1B) | **0.26** | 4.45 | **0.52** | 9.60 | **1.08** | 20.37 | **2.08** | 58.64 | **1.52** | 122.06 |
| Gated DeltaNet | 0.48 | 4.52 | 0.95 | 9.04 | 1.90 | 18.07 | 3.79 | 36.14 | 1.91 | 71.66 |
| Mamba-2 | 0.48 | 4.62 | 0.96 | 9.24 | 1.91 | 18.48 | 3.81 | 36.94 | 1.92 | 57.90 |
| Mamba-3 (SISO) | 0.48 | **4.36** | 0.95 | **8.70** | 1.90 | **17.41** | 3.80 | **34.83** | 1.91 | **54.79** |
| Mamba-3 (MIMO r=4) | 0.58 | 4.70 | 1.15 | 9.40 | 2.30 | 18.82 | 4.55 | 37.57 | 2.36 | 62.88 |

We observe that (i) Mamba-3 adds minimal forward-pass cost, showing that the exponential-trapezoidal update, complex state tracking, and MIMO parameterization remain lightweight; (ii) decode latency is competitive across recurrent models; and (iii) recurrent mixers scale more gently with context length than vLLM Llama-3.2-1B, which grows much faster with $L$ due to KV-cache overhead.

