# OpenReview forum: "Mamba-3: Improved Sequence Modeling using State Space Principles"
_ICLR.cc/2026/Conference — ICLR 2026 Oral_

### Official Review · Reviewer_2tn7 · 2025-10-31

**Soundness:** 3
**Presentation:** 3
**Contribution:** 2
**Rating:** 6
**Confidence:** 4

**Summary:**

This paper proposes Mamba-3, an extension of Mamba-2 with a more inference-focused architecture, which introduces three core improvements to its state-space model (SSM) formulation. First, it replaces Euler discretization with a trapezoidal rule, which can offer offers a lower sequence approximation error.  Second, Mamba-3 empowers a complex-valued SSM via a RoPE-like parameterization. The authors actually show the equivalence of complex-valued SSMs to real SSMs with data-dependent rotations on B and C (analogous to keys and values). Third, Mamba-3 introduces a MIMO (multiple-input multiple-output) SSMs that replace single-vector outer products with matrix-matrix updates in order to increase the arithmetic intensity during decoding. The architecture itself includes other modifications, such as QK-style normalization after B and C projections, and removes the short convolution. The paper evaluates 180M-1.5B models trained on 100B FineWeb-Edu tokens (2K context) across language modeling, retrieval (NIAH), formal languages (parity, modular arithmetic), and inference latency. Mamba-3 generally matches or outperforms Mamba-2 and Gated DeltaNet, with notable gains on state-tracking tasks and inference efficiency, particularly with MIMO.

**Strengths:**

- The three modifications are grounded in classical SSM theory and integrate cleanly into Mamba-2's framework. The generalized trapezoid recurrence (Eq. 3–4) elegantly recovers Mamba-2 as a special case ($λ_t=1$).

- Propositions 2-4 clearly derive how complex SSMs reduce to real SSMs with RoPE-style rotations. This is a clean way to say that "we can have complex dynamics at only RoPE's computational cost."

- The arithmetic intensity analysis (SISO =  $\theta(1)$, MIMO = $\theta(r)$) convincingly motivates MIMO for IO-bound decoding, and the matrix-matrix update is of course implementation-friendly.

- The ablation study in Table 4 is quite valuable. It shows that convolutions become optional is. In addition, tthe formal-language ablation where disabling RoPE makes the model underperform directly validates the complex/RoPE mechanism. This is insightful and may open new future research ideas into incorporating other forms of positional encoding into SSMs.

- Training four model sizes (180M–1.5B) on identical 100B token budgets enables fair cross-model comparisons.

- Near-perfect accuracy on parity (~100%) and modular arithmetic (~98%), plus improvements on bracketed tasks, demonstrate that Mamba-3 successfully addresses state-tracking limitations of Mamba-2.

**Weaknesses:**

- All pretraining uses 2K context, and long-sequence tests mostly stay at 2K (NIAH) or only reach 4K (2x extrapolation). For a model claiming better sequence approximation with the trapezoidal rule, evaluation should stress-test extrapolation at longer scenarios (e.g., 32K), especially since the paper notes performance drops where Gated DeltaNet sometimes wins.

- Table 3 reports single-point latencies without variance. Critically, the authors use "custom kernels" for Mamba-3 but "standard reference" implementations for baselines—making results difficult to verify. Optimized kernels for competitors could reverse the rankings. The paper must specify kernel implementations, fusion levels, and hardware details. Without this, is very hard to access the real efficiency contributions of Mamba-3.

- Figure 3's missing Mamba-3 MIMO point makes the work seem a bit sloppy. Of course, I think the model will perform well $d_{state}=32$, but still find this unconfortable.

- While the main LM results use SISO, the inference analysis argues for MIMO. In the formal language tables there's only a mention to "Mamba-3.", so it is unclear which method was used here. Since MIMO affects FLOPs, parameters, and stability, every benchmark should explicitly state "SISO" or "MIMO (rank=r)."

- For an "inference-first" paper, comparisons should include production-grade transformer inference (FlashDecoding, paged KV cache, vLLM chunked prefill, etc) at realistic batch sizes and sequence lengths. Moreover, only decoding latency is reported, despite prefill being an extremelly important part of inference in LLMs. This omission hides important costs.

- Proposition 1 claims $O(\Delta_t^2)$ global error with data-dependent $\lambda_t$ but doesn't state required regularity conditions (e.g., Lipschitz continuity).

- I found the paper to be very condensed. It seems like the distance between letters was reduced to make more things fit into the paper. This make things hard to read.

**Questions:**

- What regularity assumptions on $\lambda_t(x_t)$ guarantee quadratic global error?

- Does the rotation-absorption argument (Eq. 18–19) hold when $A_t$ is diagonal rather than scalar?

- If $\lambda_t = \frac{1}{2}$, the scheme is 2nd order; if $\lambda_t$ is data-dependent but Lipschitz and satisfies $\lambda_t = \frac{1}{2} + O(\Delta_t)$, it is still 2nd order; otherwise the order collapses to 1---Euler's rule. Is this collapse ensured in Mamba-3?

- Inference implementation details: For Table 3, specify: (a) kernel type (Triton/CUDA) and fusion level for each baseline, (b) exact configuration (batch size, sequence length, heads, dstate), (c) whether all models use comparable optimization levels.

- Did you try evaluating the 1.5B model (trained at 2K) on 8K–32K retrieval tasks, not just LM perplexity extrapolation (Fig. 5)? The LM curve is nice, but the claim of better sequence approximation would be more convincing on long retrieval where the state is actually stressed.

- Since your goal is inference improvements, could you report prefill (1K-4K) and decode (1 token) wallclock-times for a strong transformer baseline using FlashDecoding (a comparable open kernel on the same hardware is also valuable)? Even if Mamba-3 still wins on decode, showing the transformer prefill margin would make the paper easier for readers.

- Hammering on this again, but why SISO for main LM runs? If MIMO is superior at equal state size (per the inference section), why use SISO for 820M/1.5B language modeling?

---

> ### Author Response · Authors · 2025-11-30
> **Response to Reviewer 2tn7 (1/4)**
>
> We greatly appreciate the reviewer for their time and effort leaving meticulous and insightful feedback and suggestions. We are glad the reviewer liked the theoretically-grounded improvements and found the ablations “quite valuable.” We hope to answer the reviewer’s questions and concerns below.
>
> > evaluation should stress-test extrapolation at longer scenarios (e.g., 32K), especially since the paper nuotes performance drops where Gated DeltaNet sometimes wins.
>
> > Did you try evaluating the 1.5B model (trained at 2K) on 8K–32K retrieval tasks, not just LM perplexity extrapolation (Fig. 5)? The LM curve is nice, but the claim of better sequence approximation would be more convincing on long retrieval where the state is actually stressed.
>
>
> We agree with the reviewer that stress-testing retrieval capabilities at longer sequence lengths (8K–32K) is an important axis of evaluation. It is worth noting that fixed-state **linear models inherently have weaker retrieval abilities than Transformers**, whose KV cache grows with sequence length [1,2,3]. This limitation is also reflected in our results, where Mamba-3 improves over Mamba-2 on retrieval (Table 2) but still trails Transformer baselines. Recent works on **hybrid SSM-Attention architectures** have emerged as a promising direction [1,2] to address this gap.
>
> Motivated by this, we trained a 1.5B hybrid Mamba-3 SISO model with a 5:1 ratio of Mamba-3 and self-attention blocks, following the same training setup as Table 1 (details in the shared response). We report results on the NIAH benchmark [4], extrapolating from the training sequence length of 2K up to 32K. **We find that the Mamba-3 hybrid model significantly improves retrieval performance, even outperforming the Transformer baseline and maintaining near-perfect accuracy on NIAH-1 up to 16K**. Our results reinforce the idea that, while pure SSMs excel at language modeling, **hybrid models augment their retrieval capabilities through attention**.
>
> Due to limited time, we are still training hybrid variants of other baselines (e.g., Gated DeltaNet), and will include these results in the final paper.
>
> | Model                                | NIAH-1 (8192) | NIAH-1 (16384) | NIAH-1 (32768) | NIAH-2 (8192) | NIAH-2 (16384) | NIAH-2 (32768) | NIAH-3 (8192) | NIAH-3 (16384) | NIAH-3 (32768) |
> |--------------------------------------|---------------|----------------|----------------|---------------|----------------|----------------|---------------|----------------|----------------|
> | Gated Deltanet 1.3B                  | 75.8          | 38.6           | 17.0           | 16.4          | 8.0            | **10.4**           | 16.0          | 7.0            | **4.6**            |
> | Mamba 2 1.3B                         | 11.4          | 7.2            | 2.2            | 5.4           | 3.0            | 2.4            | 9.2           | 2.4            | 1.4            |
> | Mamba 3 1.3B                         | 31.2          | 13.6           | 7.4            | 13.2          | 7.8            | 7.2            | 12.2          | 4.2            | 2.4            |
> | Mamba 3 1.3B Hybrid w/ SP | **98.8**          | **99.0**           | **32.2**           | **54.8**          | **14.6**           | 2.8            | **36.8**          | **9.8**            | 0.8            |
>
> | Model                                | NIAH-1 (1024) | NIAH-1 (2048) | NIAH-1 (4096) | NIAH-2 (1024) | NIAH-2 (2048) | NIAH-2 (4096) | NIAH-3 (1024) | NIAH-3 (2048) | NIAH-3 (4096) |
> |--------------------------------------|---------------|---------------|---------------|---------------|---------------|---------------|---------------|---------------|---------------|
> | Mamba 3 1.3B Hybrid w/ SP | 100.0         | 100.0         | 99.0          | 100.0         | 99.8          | 96.8          | 99.0          | 89.4          | 83.2          |
>
> Our hybrid model was post-trained with state-passing (SP) [5] for only 100M tokens to enable better length extrapolation, as NoPE (no RoPE) self-attention layers have been shown in past work to need additional interventions to enable out-of-distribution generalization [6,7,8].

---

> ### Author Response · Authors · 2025-11-30
> **Response to Reviewer 2tn7 (2/4)**
>
> > the authors use "custom kernels" for Mamba-3 but "standard reference" implementations for baselines—making results difficult to verify. Optimized kernels for competitors could reverse the rankings. The paper must specify kernel implementations, fusion levels, and hardware details. Without this, is very hard to access the real efficiency contributions of Mamba-3.
>
> We appreciate the reviewer raising this point and fully agree with the need for more precise framing of our kernel contributions and rigor in latency reporting. We would like to take this opportunity to clarify the intended message of Table 3, by which we wanted to demonstrate the following:
>
> 1. The added components in Mamba-3----trapezoidal rule, complex update, and MIMO----are **lightweight** and do not introduce meaningful latency overhead. Our intention was **not** to claim that Mamba-3 is *inherently* faster than Mamba-2. Rather, since the Mamba-3 kernel strictly generalizes the Mamba-2 kernel, a fully optimized Mamba-2 kernel should be slightly faster. However, **the expected difference is minimal**, as the added components do not significantly impact latency.
> 2. Optimized Mamba-3 kernels are a **core part** of the paper's contribution. We implemented the Mamba-3 decode kernel in CuTe DSL, which was made significantly easier by the **simplicity** of Mamba-3 components, e.g., the complex update uses the RoPE trick, MIMO is implemented via fast matrix multiplications. We believe releasing optimized kernels helps the community adopt the models more easily by removing the need for low-level optimization. This is why we evaluated against the **fastest baseline kernels** released by the respective model developers, [fla](https://github.com/fla-org/flash-linear-attention) and [mamba-ssm](https://github.com/state-spaces/mamba). We believe that releasing fast, ready-to-use kernels enhances the practical value of our contribution.
> --------
> > Inference implementation details: For Table 3, specify: (a) kernel type (Triton/CUDA) and fusion level for each baseline, (b) exact configuration (batch size, sequence length, heads, dstate), \(c) whether all models use comparable optimization levels.
>
> > For an "inference-first" paper, comparisons should include production-grade transformer inference (FlashDecoding, paged KV cache, vLLM chunked prefill, etc) at realistic batch sizes and sequence lengths. Moreover, only decoding latency is reported, despite prefill being an extremelly important part of inference in LLMs. This omission hides important costs.
>
> > Since your goal is inference improvements, could you report prefill (1K-4K) and decode (1 token) wallclock-times for a strong transformer baseline using FlashDecoding (a comparable open kernel on the same hardware is also valuable)? Even if Mamba-3 still wins on decode, showing the transformer prefill margin would make the paper easier for readers.
>
> The reviewer is absolutely right that implementation details should be made explicit, and we apologize for this oversight. We provide a table detailing the kernel type, fusion level, hardware setup, and architectural configuration for each baseline, along with the mean, std deviation of prefill and prefill+decode latency.
>
> ### Forward Kernels
>
> | Model  | DSL | Fusion Level                                                                                               |
> |-------------------|-----------:|------------------------------------------------------------------------------------------------------------|
> | Mamba-2           | Triton     | Input Projection (Torch), 1D-Conv (Triton), SSM (Triton), Gate and Output Projection (Torch)              |
> | Gated DeltaNet    | Triton     | Input Projection (Torch), 1D-Conv (Triton), Chunked Delta Rule (Triton), Gate and Output Projection (Torch) |
> | Mamba-3 (SISO)    | Triton     | Input Projection (Torch), SSM (Triton; fused with Rotary), Gate and Output Projection (Torch)            |
> | Mamba-3 (MIMO)    | Triton     | Input Projection (Torch), SSM (Triton; fused with Rotary), Gate and Output Projection (Torch)            |
>
> ### Decode Kernels
>
> | Model    | DSL      | Fusion Level                                                                                               |
> |-------------------|-----------------|--------------------------------------------------------------------------------------------------------------|
> | Mamba-2           | Triton          | Input Projection (Torch), 1D-Conv (Triton), SSM (Triton), Gate and Output Projection (Torch)                |
> | Gated DeltaNet    | Triton          | Input Projection (Torch), 1D-Conv (Triton), Recurrent Delta Rule (Triton), Gate and Output Projection (Torch) |
> | Mamba-3 (SISO)    | CuTe + Triton   | Input Projection (Torch), Rotary (Triton), SSM (CuTe; fused with Gate and Output Projection)                |
> | Mamba-3 (MIMO)    | CuTe + Triton   | Input Projection (Torch), Rotary (Triton), SSM (CuTe; fused with Gate and Output Projection)                |

---

> ### Author Response · Authors · 2025-11-30
> **Response to Reviewer 2tn7 (3/4)**
>
> **Models.** We measured prefill and prefill+decode latency for Mamba-3 1.5B, Mamba-2, Gated DeltaNet (GDN), and using the vLLM engine (v0.11.0) for [Llama-3.2 1B](https://huggingface.co/meta-llama/Llama-3.2-1B). Mamba-3, Mamba-2, and GDN were evaluated at the 1.5B scale with `model dimension = 2048`, `number of layers = 24`. For Mamba models: `state size = 128`, `head dimension = 64`; for GDN: `qk_headdim = 128` and `v_expand = 2`.
>
> **Setting.** Sequence lengths were swept over {512, 1k, 2k, 4k, 16k} for prefill, with an equal number of tokens decoded. For sequence lengths {512, 1k, 2k, 4k}, we use batch size of 128; for sequence lengths {16k}, we use batch size of 16. All experiments were run on a single H100-SXM 80GB GPU.
>
> ### Prefill Latency
>
> | Model                         | 512          | 1024         | 2048         | 4096        | 16384       |
> |-------------------------------|--------------|--------------|--------------|-------------|-------------|
> | vLLM (Llama-3.2-1B)           | **0.26±0.01** | **0.52±0.00** | **1.08±0.02** | **2.08±0.04** | **1.52±0.03** |
> | Gated DeltaNet                | 0.48±0.00     | 0.95±0.00     | 1.90±0.00     | 3.79±0.00    | 1.91±0.00    |
> | Mamba-2                       | 0.48±0.00     | 0.96±0.00     | 1.91±0.00     | 3.81±0.01    | 1.92±0.01    |
> | Mamba-3 (SISO)                | 0.48±0.00     | 0.95±0.00     | 1.90±0.00     | 3.80±0.00    | 1.91±0.00    |
>
> ### Prefill+Decode Latency
>
> | Model                         | 512          | 1024         | 2048         | 4096        | 16384       |
> |-------------------------------|--------------|--------------|--------------|-------------|-------------|
> | vLLM (Llama-3.2-1B)           | 4.45±0.12     | 9.60±0.16     | 20.37±0.25    | 58.64±1.23   | 122.06±0.08  |
> | Gated DeltaNet                | 4.52±0.01     | 9.04±0.02     | 18.07±0.03    | 36.14±0.06   | 71.66±0.23   |
> | Mamba-2                       | 4.62±0.01     | 9.24±0.01     | 18.48±0.03    | 36.94±0.06   | 57.90±0.23   |
> | Mamba-3 (SISO)                | **4.33±0.01** | **8.64±0.01** | **17.29±0.03** | **34.57±0.06** | **53.97±0.28** |
>
> From our latency analysis, we observe that
> 1.  **Mamba-3 SISO achieves the *fastest* prefill+decode latency at all sequence lengths**, outperforming Mamba-2, Gated DeltaNet and even the highly optimized vLLM LLaMA-3.2-1B baseline----despite the latter benefiting from mature, hand-tuned kernels and systems engineering.
> 2. Prefill latencies across Mamba-2 and Mamba-3 are nearly identical, confirming that Mamba-3’s added components (e.g., RoPE, MIMO) do not introduce forward-pass overhead.
>
> We would like to emphasize that production-grade Transformer inference engines, like vLLM, have had years to mature, while the original Mamba paper was only released two years ago. While Mamba-3's speeds are already competitive, we expect prefill and decode speeds to continue to improve as the Mamba ecosystem continues to mature.
>
> ---------
>
>
> > Figure 3's missing Mamba-3 MIMO point makes the work seem a bit sloppy. Of course, I think the model will perform well d_state=32, but still find this unconfortable.
>
> We have added a new [Figure 3](https://i.postimg.cc/MXtx34XC/pareto-mamba.png) in our manuscript which directly compares Mamba-2, Mamba-3 SISO, and Mamba-3 MIMO across $d_\text{state}=\{16,32,64,128\}$. All models are controlled for state size and parameters. Here we continue to find that **Mamba-3 MIMO is pareto-optimal for state usage at all sizes**.
>
> Our [original Figure 3 (with GDN baseline)](https://i.postimg.cc/V541ZT5K/pareto-gdn.png) has been updated with the missing point. We have moved it to the Appendix due to the difficulty controlling the architecture of GDN vs Mamba-style models due to varying head structures (multi-head vs multi-value for GDN vs Mamba). We discuss this in detail in our Appendix F, Line 1184.
>
> -------------------
>
> > In the formal language tables there's only a mention to "Mamba-3.", so it is unclear which method was used here. Since MIMO affects FLOPs, parameters, and stability, every benchmark should explicitly state "SISO" or "MIMO (rank=r)."
>
> We apologize for any confusion this may have caused. In our paper, when we refer to Mamba-3, it is the SISO variant unless explicitly stated as MIMO, with all MIMO experiments run with rank $r=4$. Since the submission, we have expanded our exploration of MIMO’s $r$ dynamics, which we have added in the general response. We will adjust our terminology in the revision to improve clarity, and thank the reviewer for their suggestion.

---

> ### Author Response · Authors · 2025-11-30
> **Response to Reviewer 2tn7 (4/4)**
>
> > Proposition 1 claims O(Delta_t^2) global error with data-dependent Lambda_t but doesn't state required regularity conditions (e.g., Lipschitz continuity).
>
> > What regularity assumptions on lambda_t(x_t) guarantee quadratic global error?
>
> > If lambda_t = 1/2, the scheme is 2nd order; if lamba_t is data-dependent but Lipschitz and satisfies lambda_t = ½ + O(Delta_t), it is still 2nd order; otherwise the order collapses to 1---Euler's rule. Is this collapse ensured in Mamba-3?
>
> We have done a deeper analysis of this result and agree with the reviewer that $\lambda_t$ must satisfy $\lambda_t = 1/2 + O(\Delta_t)$ to theoretically guarantee second-order convergence. **We are grateful to the reviewer for pointing this out; We have amended the Remark 2 (Line 189) statement in the paper to reflect this condition as well as added the assumptions and proof in Appendix B (Line 843).**
>
> Below we show ablations on variants of the parametrization.
> **Setting:** All runs use the Mamba-3 (SISO) 440M model trained at Chinchilla scale, with the other architectural and optimization hyperparameters being the same as in Language Modeling experiments.
>
>
> | Lambda Parameterization        | Pretraining Validation PPL |
> |--------------------------------|----------------------------|
> | Default                        | 15.72                      |
> | Fixed 1/2                      | 15.76                      |
> | No trap                        | 15.81                      |
>
> The default model uses a data-dependent gate $\lambda_t = \sigma(u_t)$, where $u_t$ is a learned projection of the current input token. We find that the default parameterization empirically performs the best. Hence we choose the simpler default parameterization that does *not* enforce the $1/2 + O(\Delta_t)$.
>
> -------------
>
> > Hammering on this again, but why SISO for main LM runs? If MIMO is superior at equal state size (per the inference section), why use SISO for 820M/1.5B language modeling?
>
> We would like to re-emphasize the core trade-off between SISO and MIMO: **MIMO offers better modeling performance without increasing inference cost, but it does increase training cost.** This is because decoding is memory-bound, while training is already compute-bound. We have revised the draft to make this more salient (Line 332).
>
> Our general response reinforces that MIMO models continue to improve upon its corresponding SISO model as model size scales. However, our main results use SISO to be a fair comparison against standard linear model baselines, which can all be viewed as SISO models. As our general response also notes, SISO Mamba-3 is as fast as popular baselines such as Mamba-2 and GDN.
>
> -----------------
>
> > Does the rotation-absorption argument (Eq. 18–19) hold when A_t is diagonal rather than scalar?
>
> Based on our Remark 3, our rotation-absorption argument holds for any $A_t$ that is a block diagonal matrix with $2\times2$ blocks that are scalar multiples of the identity, i.e., $A_t = D \otimes I_2)$ where $D$ is a diagonal matrix.
>
> -----------------
>
> > I found the paper to be very condensed. It seems like the distance between letters was reduced to make more things fit into the paper. This make things hard to read.
>
> We apologize for any challenges faced while reading our paper. We hope the additional page allotted after discussions may help alleviate some of these issues.
>
> ----------------------------
>
> ## References
>
> [1] [2402.18668] Simple linear attention language models balance the recall-throughput tradeoff
>
> [2] [2406.07887] An Empirical Study of Mamba-based Language Models
>
> [3] https://qwen.ai/blog?id=4074cca80393150c248e508aa62983f9cb7d27cd
>
> [4] [2404.06654] RULER: What's the Real Context Size of Your Long-Context Language Models?
>
> [5] [2507.02782] Understanding and Improving Length Generalization in Recurrent Models
>
> [6] [2501.18795] Rope to Nope and Back Again: A New Hybrid Attention Strategy
>
> [7] [2504.08719] SWAN-GPT: An Efficient and Scalable Approach for Long-Context Language Modeling
>
> [8] [2404.12224] Length Generalization of Causal Transformers without Position Encoding

---

> > ### Comment · Reviewer_2tn7 · 2026-02-26
> > **Follow up**
> >
> > It is a pity that we were not able to discuss this further during the rebuttal period. I had to re-read the paper and the entire discussion to refresh my memory on this paper. In any case, I thank the authors for their meticulous work and for the additional experiments. I have two additional questions.
> >
> > - Do you also have latency numbers for Mamba-3 MIMO?
> >
> > - Are the inference speedups dependent on Hopper-specific kernel features, or do they persist on Ampere GPUs as well?
> >
> > And two general comments:
> > - So, it turns out that the default learned gate **does not** enforce the condition needed for second-order convergence. I think it would be important to emphasize that in the main paper and point to the appendix for the ablation results (to justify the choice).
> >
> > - Thank you for running the long-context retrieval experiments. The results are quite interesting. So, the evidence points more to a hybrid setup for long-context modeling. This is, of course, a minor point, but I strongly believe these results would complement the final version of the paper well, and it would also be great to include a few words on their implications.

---

### Official Review · Reviewer_h5BJ · 2025-11-01

**Soundness:** 2
**Presentation:** 3
**Contribution:** 2
**Rating:** 6
**Confidence:** 5

**Summary:**

Mamba-3 is a state-space model (SSM) designed from an inference-first perspective that addresses the limitations of existing sub-quadratic sequence models (e.g., Mamba-2, Gated DeltaNet) in quality, state-tracking capability, and hardware efficiency. It achieves SOTA performance across language modeling, state-tracking, and retrieval tasks while maintaining linear compute and constant memory, setting a new Pareto frontier for performance-efficiency tradeoffs.

**Strengths:**

1. The three core improvements, trapezoidal discretization, complex-valued SSMs, and MIMO formulation, are not isolated tweaks but creative combinations of classical SSM theory and modern LLM needs. Trapezoidal discretization generalizes Euler’s rule (used in Mamba-2) to a second-order accurate recurrence, while the complex-valued SSM recovers rotational dynamics absent in real-valued counterparts.

2. Rigorous Theory & Comprehensive Empirical Validation. The paper includes rigorous proofs for key propositions (e.g., complex-to-real SSM equivalence, trapezoidal discretization error bounds) and formalizes the mathematical underpinnings of each innovation. This theoretical rigor ensures the model’s behavior is well-understood, not just empirically effective.

3. The paper reports detailed experimental settings (training data, tokenization, hyperparameters) and provides clear metrics (perplexity, accuracy, TPS, arithmetic intensity) that enable follow-up work.
And structured presentation & accessible exposition increses the avaliability.

4. Addressing Critical LLM Deployment Pain Points. By pushing the Pareto frontier of performance vs. inference efficiency, Mamba-3 directly addresses a key bottleneck in LLM deployment—scaling inference without prohibitive memory/compute costs. The MIMO variant, in particular, improves hardware utilization without increasing state size, making it viable for real-world applications.

**Weaknesses:**

1. Inherent limitations of retrieval capabilities and insufficient comparison. Mamba-3 significantly lags behind Transformer in information extraction tasks from semi-structured/unstructured data, and the root causes and potential solutions are not explored in depth.

Table 2 of the paper shows that Mamba-3 with 1.5B parameters has poor accuracy in real-world retrieval tasks such as SWDE and FDA; even in the synthetic "needle-in-a-haystack" task, when the context length exceeds 2048, the accuracy of Mamba-3 is still far below the theoretical upper limit of Transformer, and it is not compared with linear models specifically optimized for retrieval (such as Arora et al., 2025b's "Just Read Twice").

2. Limited experimental scale of the key variant (MIMO) and incomplete hyperparameter ablation. The MIMO variant was only trained on a 440M parameter model (paper 4.2.3). Because the "computational constraint" was not extended to a 1.5B scale, its performance-efficiency tradeoff under larger models could not be verified. Furthermore, the MIMO rank r was fixed at 4, and the impact of different values ​​such as r=2 and 8 on arithmetic strength and inference speed was not tested (Figure 2 only shows the theoretical trend of r, without experimental data).

3. The paper's baseline only includes Mamba-2, Gated DeltaNet, and the standard Transformer, omitting similar linear models such as Rwkv-7, DeltaProduct, and Block-Biased Mamba. These models all innovated on the expressiveness or efficiency of linear models, and some performed exceptionally well on state tracking tasks.

**Questions:**

1. Complex-Valued SSM & RoPE Equivalence: The paper argues that complex-valued SSMs are equivalent to data-dependent RoPE embeddings and compute-efficient. Could you provide a direct comparison between complex-valued SSMs and *explicitly applying RoPE* (e.g., on B/C projections of Mamba-2) in terms of: (1) training/inference speed, (2) memory usage, and (3) state-tracking performance?

2. MIMO Rank Selection: The MIMO variant uses rank \( r=4 \) for 440M models, but there is no analysis of how \( r \) impacts performance-efficiency tradeoffs. What is the marginal gain of increasing \( r \) (e.g., \( r=8, 16 \))? At what point does \( r \) become compute-bound (negating efficiency gains)? Additionally, why was \( r=4 \) chosen as the default, was it optimized via grid search or heuristic?




3. Ablations. The BC bias synergizes with trapezoidal discretization, but you do not specify its initialization strategy (e.g., zero, Xavier, data-dependent) or whether it is updated during training. Does the bias’s initialization significantly affect performance?

---

> ### Author Response · Authors · 2025-11-30
> **Response to Reviewer h5BJ (1/3)**
>
> We thank the reviewer for the time and effort spent understanding the paper and leaving insightful comments and suggestions. We are glad they found the three core combinations “creative” and liked our “rigorous theory”, “comprehensive empirical validation”, and “detailed experimental settings.” The reviewer raises some insightful questions regarding certain design aspects and ablations which we hope to answer below.
>
> > Mamba-3 significantly lags behind Transformer in information extraction tasks from semi-structured/unstructured data, and the root causes and potential solutions are not explored in depth.
>
> **We believe that linear layers, such as Mamba-3, will be most effective in hybrid attention-SSM architectures**, as they alleviate both linear model retrieval limitations and outperform pure attention-based models [1,2]. Consequently, we compare the performance of our Mamba-3 SISO hybrid to the Transformer baseline on real-world retrieval tasks in our shared response. We would like to highlight, on information extraction tasks on semi-structured and unstructured data, that our hybrid model is able to **outperform the Transformer baseline by nearly 10 percentage points and 7.5 percentage points on SWDE and FDA respectively.**
>
> ------------------------
>
> > it is not compared with linear models specifically optimized for retrieval
>
> Arora et al. (2025b) introduce two modular interventions that can be applied to any decoder-only LLM: (i) JRT-RNN, a bidirectional encoder paired with a causal decoder, and (ii) a context-repeating strategy that enables a second pass over the prompt. We view both interventions as complementary to our work----Mamba-3 can serve as a drop-in replacement for the linear-attention components used in the paper. While benchmarking these combinations is an exciting direction for future work, we consider it outside the scope of this submission, which focuses on improving the underlying backbone itself.
>
> --------------------------
>
> > Limited experimental scale of the key variant (MIMO) and incomplete hyperparameter ablation. The MIMO variant was only trained on a 440M parameter model (paper 4.2.3). Because the "computational constraint" was not extended to a 1.5B scale, its performance-efficiency tradeoff under larger models could not be verified.
>
> We have trained a 820M MIMO Mamba-3 and report the performance in the shared response. We would like to highlight that our MIMO variant continues to **outperform both the SISO variant and other baselines by nearly 1 percentage point** on averaged downstream language modeling tasks. We are currently training a 1.5B MIMO variant and will update the manuscript once finished.
>
> ----------------------
>
> > the MIMO rank r was fixed at 4, and the impact of different values such as r=2 and 8 on arithmetic strength and inference speed was not tested (Figure 2 only shows the theoretical trend of r, without experimental data).
>
> > What is the marginal gain of increasing ( r ) (e.g., ( r=8, 16 ))? At what point does ( r ) become compute-bound (negating efficiency gains)? Additionally, why was ( r=4 ) chosen as the default, was it optimized via grid search or heuristic?
>
> We measure the wall clock time and bandwidth of our decode/step function with varying MIMO rank $r$ using $d_\text{state}=128$ and fp32 tensors.
>
>
> | MIMO Rank | Wallclock time (μs) | Memory Bandwidth (GB/s) |
> |----------|----------------------|-------------------------|
> | 1 (SISO) | 239                  | 2286                    |
> | 2        | 255                  | 2156                    |
> | 4        | 260                  | 2145                    |
> | 8        | 283                  | 2030                    |
>
>
> Increasing MIMO rank only slightly decreases memory bandwidth and results in a sublinear increase in latency. Additionally , we observe that even with $r=8$, our kernel has high memory bandwidth, which indicates there is room to optimize the kernel further.
>
> The marginal gain of increasing $r$ is increased decoding FLOPs without any change to hidden state size and minimal disruption to latency. In turn, the increased decoding FLOPs, as we’ve shown both in our SISO vs MIMO results and the MIMO ablations in the shared response, translate to better performance.
>
> The point at which $r$ becomes compute-bound depends on the ops-to-byte ratio of the hardware. For instance, if we do not use tensor cores for matmul and set all variables to be of type fp32, then the ratio would be 62 TFLOPS of fp32 / 3.35TB/s ~= 18 fp32 ops per byte (for NVIDIA H100-SXM5 with respect to the DRAM). Hence, given the MIMO arithmetic intensity of $\sim r$ (for 4-byte data), the tipping point between a memory- and compute-bound kernel occurs when $r \sim 18$ (ignoring the latency with respect to the SRAM). On the other hand, one could utilize tensor cores by setting vars to be bfloat16, which would increase the ratio to ~300 bf16 ops per byte and a tipping point of $r\sim150$.

---

> ### Author Response · Authors · 2025-11-30
> **Response to Reviewer h5BJ (2/3)**
>
> We initially selected MIMO $r=4$ as a reasonable value given our compute constraints. In particular, it was not chosen based on model performance, although we do find that increasing $r$ improves performance as exhibited in our shared response.
>
> -------------------------
>
> > The paper's baseline only includes Mamba-2, Gated DeltaNet, and the standard Transformer, omitting similar linear models such as Rwkv-7, DeltaProduct, and Block-Biased Mamba.
>
> Our baseline selection was constrained by compute, so we prioritized the most widely used architectures: Gated DeltaNet, Mamba-2, and Transformers. Both Mamba-2, which also serves as the canonical predecessor baseline, and Gated DeltaNet have been validated at scale [2,3].
>
> We report the comparisons to a Block-Biased Mamba-2 below. Models were trained at the 440M Chinchilla token scale. We find that both B and C biases are required to enable strong performance.
>
> | B Bias | C Bias | Pretraining Val PPL |
> |--------|--------|---------------------|
> | No     | No     | 16.52              |
> | Yes    | No     | 16.68 (Block-Biased)|
> | No     | Yes    | 15.98              |
> | Yes    | Yes    | **15.69** (Mamba-3)|
>
> -------------------
>
> > The BC bias synergizes with trapezoidal discretization, but you do not specify its initialization strategy (e.g., zero, Xavier, data-dependent) or whether it is updated during training. Does the bias’s initialization significantly affect performance?
>
> We apologize for not explicitly mentioning the BC bias initialization strategy. Mamba-3’s B and C biases are trainable, data independent parameters initialized to all-ones, and we have added this clarification into our paper on Line 345. We train 440M Mamba-3 models to 1x Chinchilla tokens while varying their B, C bias parameterization and report the pretrain validation perplexity below. We find that our model is not very sensitive to initialization as long as some biases are initialized to positive values, consistent with our bias hypothesis, which is that the B,C bias enables the model to run a linear time-invariant and data-dependent SSM simultaneously. We chose a bias initialization of all-ones due to its simplicity.
>
> Both tables explore B and C biases have been also added to the Appendix.
>
> | B,C Bias Initialization | Trainable | Pretraining Val PPL |
> |-------------------------|-----------|----------------------|
> | 1.0 (Default)           | Yes       | 15.72                |
> | 0.0                     | Yes       | 16.57                |
> | 1.0                     | No        | 15.80                |
> | U[0,1]                  | Yes       | 15.76                |
> | U[-1,1]                 | Yes       | 16.07                |
>
> ------------------
>
> > The paper argues that complex-valued SSMs are equivalent to data-dependent RoPE embeddings and compute-efficient. Could you provide a direct comparison between complex-valued SSMs and explicitly applying RoPE (e.g., on B/C projections of Mamba-2) in terms of: (1) training/inference speed, (2) memory usage, and (3) state-tracking performance?
>
> To our understanding, the reviewer is asking about the difference between complex-valued SSMs (data-dependent RoPE) and explicitly applying vanilla (data-independent) RoPE.
> - When directly comparing the training speed of our data-dependent RoPE vs vanilla RoPE, most of the speed overhead comes from the cumulative calculation of angles. We can calculate this efficiently, resulting in the final data-dependent RoPE operation only consisting of around 3-4% of the final layer’s wall clock time. Inference speed will be comparable to default RoPE.
> - Our data-dependent RoPE does increase memory usage during training as one must first calculate the angles applied at each timestep, resulting in a $\mathcal{O}(BLHN)$ increase in memory usage per layer where $B$ is the batch size, $L$ is the sequence length, $H$ is the number of heads, and $N$ is the $d_\text{state}$. This however reduces down to $\mathcal{O}(BHN)$ during decode as only the final cumulative angle is required. This is dominated by the memory required for the state, which is $\mathcal{O}(BHNP)$ where $P$ is the headdim.
> - We find that applying default RoPE to our Mamba-3 layer degrades state-tracking performance back random guessing for parity and arithmetic with brackets and degrades arithmetic without brackets significantly. We report the new numbers in our updated Table 4, and paste the table below for ease of reference.

---

> ### Author Response · Authors · 2025-11-30
> **Response to Reviewer h5BJ (3/3)**
>
> | Model                      | Parity | Arith. w/o brackets | Arith. w/ brackets |
> |---------------------------|----------|------------------------|----------------------|
> | Mamba-3                   | 100.00   | 98.51                  | 87.75                |
> | Mamba-3 (w/o RoPE)        | 2.27     | 1.49                   | 0.72                 |
> | Mamba-3 (w/ Std. RoPE)    | 1.56     | 20.70                  | 2.62                 |
> | Mamba-2                   | 0.90     | 47.81                  | 0.88                 |
> | Gated DeltaNet [-1,1]     | 100.00   | 99.25                  | 93.50                |
>
> ---------------------------
> ## References
>
> [1] [2402.18668] Simple linear attention language models balance the recall-throughput tradeoff
>
> [2] [2406.07887] An Empirical Study of Mamba-based Language Models
>
> [3] https://qwen.ai/blog?id=4074cca80393150c248e508aa62983f9cb7d27cd
>
> [4] [2506.02475] Comba: Improving Bilinear RNNs with Closed-loop Control

---

### Official Review · Reviewer_5Dn9 · 2025-11-01

**Soundness:** 3
**Presentation:** 3
**Contribution:** 3
**Rating:** 8
**Confidence:** 4

**Summary:**

Current State-Space Models have some draw-backs compared to Transformer based models, such as lack certain capabilities, low generalization quality, not hardware-efficient, etc. This paper proposes the Mamba-3 architecture, from inference-first perspective. Specifically, the trapezoidal discretization is used to improve the expressive dynamics, complex-valued state spaces is introduced for state-tracking, and the multi-input multi-output (MIMO) mechanism is proposed for hardware efficiency. Experiments of show that Mamba-3 outperforms Mamba-2 or even transformer based models across 180M to 1.5B parameters.

**Strengths:**

1. The proposed Generalized Trapezoidal Discretization is more accurate than the Euler's rule used in Mamba2, by using the second-order  approximation of the integral.
2. The complex SSM is novel, which is equivalent to a real SSM with data-dependent rotary embeddings (RoPE). Detailed theoretical analyses are provided.
3. The MIMO method efficiently solves the I/O problem, which is a key bottleneck for Mamba-2.

**Weaknesses:**

More complex tasks such as reasoning can be explored to fully demonstrate the capability of Mamba-3.

**Questions:**

Can the proposed Mamba-3 be generalized to vision tasks?

Does Mamba-3 still has advantages when compared to sparse attention based transformers?

---

> ### Author Response · Authors · 2025-11-30
> **Response to Reviewer 5Dn9 (1/1)**
>
> We are grateful for the reviewer’s highly positive evaluation of our work. We are glad they viewed the three core components of Mamba-3 as strengths and liked the theoretical analysis provided for our complex SSM. We hope to answer the questions posed by the reviewer below.
>
> > More complex tasks such as reasoning can be explored to fully demonstrate the capability of Mamba-3.
>
> We evaluated our 1.5B models and baselines on five additional benchmarks: BoolQ, CommonsenseQA, and MMLU (continuation variant), SciQ, and LogiQA2. We find that our **Mamba-3 SISO model continues to perform well on multiple benchmarks with the hybrid variant further improving performance**. Unfortunately, more complex tasks, such as mathematical reasoning, often require increased pre-training plus post-training. Thus, we leave such tasks to subsequent work. We hope the increased suite of evaluations helps further show the full capabilities of Mamba-3.
>
> | Model               | BoolQ | CommonSense QA | MMLU | SciQ | LogiQA2 (Acc_n) |
> |---------------------|-------|----------------|------|------|------------------|
> | Llama               | 59.9  | 19.7           | 34.0 | 92.1 | **29.8**             |
> | Gated DeltaNet      | 59.8  | 19.7           | **34.9** | 90.7 | 27.7             |
> | Mamba 2             | 62.0  | 20.7           | **34.9** | 91.0 | 27.5             |
> | Mamba 3 SISO        | 60.6  | 21.3           | **34.9** | 91.0 | 28.2             |
> | Mamba 3 Hybrid SISO | **64.1**  | **22.5**           | 34.5 | **92.5** | 28.5             |
>
> -------------------------
>
> > Can the proposed Mamba-3 be generalized to vision tasks?
>
> As Mamba-3 is a general sequence mixer, it can be generalized to vision tasks and other modalities, much like how its predecessors Mamba-1 and Mamba-2 have been [1,2,3].
>
> ------------------
>
> > Does Mamba-3 still has advantages when compared to sparse attention based transformers?
>
> While sparse attention is viewed as an efficient but worse-performing approximation to quadratic attention, we believe that linear models such as state-space models and linear attention have distinct performance trade-offs compared to full attention: they are not only more efficient but can have better modeling performance in many scenarios. This is exemplified by the performance on other modalities [4], as well as the fact that hybrid linear and attention models are known to have better performance than a full Transformer [5]. For this reason, hybrid SSM/Linear+Transformer models have become increasingly popular as frontier models [6,7,8,9]
>
> ---------
> ## References
>
> [1] [2401.09417] Vision Mamba: Efficient Visual Representation Learning with Bidirectional State Space Model
>
> [2] [2403.06977] VideoMamba: State Space Model for Efficient Video Understanding
>
> [3] [2510.12111] Chimera: State Space Models Beyond Sequences
>
> [4] [2403.03234] Caduceus: Bi-Directional Equivariant Long-Range DNA Sequence Modeling
>
> [5] [2406.07887] An Empirical Study of Mamba-based Language Models
>
> [6] [2505.00949] Llama-Nemotron: Efficient Reasoning Models
>
> [7] [2505.15431] Hunyuan-TurboS: Advancing Large Language Models through Mamba-Transformer Synergy and Adaptive Chain-of-Thought
>
> [8] https://qwen.ai/blog?id=4074cca80393150c248e508aa62983f9cb7d27cd
>
> [9] [2510.26692] Kimi Linear: An Expressive, Efficient Attention Architecture

---

### Official Review · Reviewer_kFu5 · 2025-11-03

**Soundness:** 3
**Presentation:** 4
**Contribution:** 3
**Rating:** 8
**Confidence:** 4

**Summary:**

This paper introduces Mamba-3, a followup sequence model architecture that builds upon the Mamba family of SSMs. The work is motivated by an "inference-first" perspective, aiming to create a model that excels across three axes: quality, capability, and inference efficiency, addressing limitations in prior sub-quadratic models like Mamba-2.  Basically, this paper propose three core, major improvements derived from classical SSM theory and also validate Mamba-3 through extensive experiments, showing that it outperforms strong baselines in language modeling, possesses new state-tracking capabilities on synthetic tasks, and establishes a new Pareto-frontier for performance versus inference cost.   It presents a clear, significant, and well-executed advancement in the design of efficient sequence models. The paper is strong for three main reasons:

1.  **Principled Improvements over existing SSMs/Mamba family:** The core contributions are not arbitrary architectural tweaks but are grounded in control theory and SSM principles. The connection drawn between complex SSMs and data-dependent RoPE is very elegant IMO and provides a clear theoretical justification for the observed gains in capability.

2.  **Holistic Improvement:** The work addresses the multi-faceted challenge of building a practical LLM. It simultaneously pushes forward model quality, unlocks new fundamental capabilities (state-tracking issue which has been rooted in RNNs), and improves hardware-level efficiency(MIMO formulation). This is a rare and impactful combination. (Though it will be more impactful if all artifacts will be open-sourced.

3.  **Compelling and Rigorous Empirical Evidence:** The experimental validation is thorough and persuasive. The head-to-head comparisons on language modeling, the stark success/failure results on formal language tasks, and the Pareto-frontier analysis collectively provide a powerful and convincing case for the superiority of the Mamba-3 design.

**Strengths:**

- **Theoretical Elegance and Insight:** The paper's standout strength is its use of classical SSM theory to motivate architectural changes. The derivation of a data-dependent RoPE from a complex state-space is a beautiful and insightful result that bridges two important lines of work.

- **Targeted Problem Solving:** Each of the three methodological changes directly targets a known weakness in prior models. Trapezoidal discretization for expressivity, complexification for state-tracking, and MIMO for hardware utilization. This focused approach makes the paper's narrative clear and its contributions easy to appreciate.

- **Excellent Empirical Validation:** The experiments are top-notch. The Pareto-frontier plot (Figure 3) is a highly effective way to visualize the trade-off between performance and inference cost, clearly showing Mamba-3's dominance. The ablation on state-tracking capabilities (Table 4b) is a decisive demonstration of the complex SSM's impact.

*   **Practicality and Inference-First Focus:** The introduction of the MIMO variant shows a deep understanding of the practical bottlenecks in LLM deployment. By focusing on arithmetic intensity, the authors address a subtle but critical aspect of real-world performance that goes beyond theoretical FLOPs.

-  **High-Quality Presentation:** The paper is extremely well-written, with complex ideas explained clearly and supported by well-designed figures and tables.

**Weaknesses:**

My major concern is that retrieval Capabilities Still Lag Transformers: While the paper is honest about this, and Mamba-3 shows improvement over Mamba-2, the results in Table 2 confirm that a fundamental gap in retrieval performance versus Transformer models remains. This is an inherent challenge for fixed-state recurrent models and represents a key limitation. Maybe authors can add some potential exploration about hybrid models? I think we are at the age that Hybrid models are becoming more important and can help alleviate the shortages of RNNs which we should acknowledge and make good use of.

**Questions:**

- The results on retrieval (Table 2) are improved but still trail the Transformer baseline significantly. Do you view this as a fundamental ceiling for fixed-state models, or do you see pathways for future SSM-based architectures to further close this gap? Maybe authors can add some potential exploration about hybrid models? I think we are at the age that Hybrid models are becoming more important and can help alleviate the shortages of RNNs which we should acknowledge and make good use of.

- The "RoPE trick" is a very efficient way to implement the complex SSM. Could you briefly comment on how the cumulative product of rotation matrices `(Π R_i)` is handled efficiently during both the parallel training scan and the sequential inference steps?

Additional questions:

- The MIMO formulation is a compelling idea for improving hardware efficiency. Could you elaborate on the selection of the MIMO rank `r=4`? Have you explored the sensitivity of model performance and inference latency to different values of `r`? Is there a risk of diminishing returns or even performance degradation if `r` becomes too large?

- The ablation in Table 4a convincingly shows that the trapezoidal discretization and BC bias make the short convolution redundant. Can you provide more intuition on *why* this is the case? Does the learned size-two convolution from the trapezoidal rule effectively capture the same local information as a dedicated convolution layer?

---

> ### Author Response · Authors · 2025-11-30
> **Response to Reviewer kFu5 (1/2)**
>
> We greatly appreciate the reviewer’s positive assessment of our work—highlighting Mamba-3 as "principled and theoretically elegant", the experiments as "top-notch", and the paper as "extremely well-written". We now address the questions and concerns raised by reviewer below:
>
> > This [retrieval capabilities] is an inherent challenge for fixed-state recurrent models and represents a key limitation. Maybe authors can add some potential exploration about hybrid models?
>
> We thank the reviewer for raising this important point----both the reviewer and our manuscript note that while Mamba-3 improves over Mamba-2 in retrieval capability, it still trails behind Transformers (Table 2). This finding aligns with prior works [1,2,3] showing that fixed, finite-state linear models have weaker retrieval abilities than Transformers, whose KV cache grows with sequence length. While this represents a fundamental limitation of finite state models, our results suggest that expressive state transition matrices, as used in Mamba-3, can meaningfully improve retrieval with better state handling. We see this as one of the pathways for future SSMs to further reduce this gap.
>
> Additionally, **we agree with the reviewer that hybrid attention-SSM architectures are a promising research direction**, as they both alleviate SSM retrieval limitations and outperform pure attention based models [1,2,3]. To explore the potential of Mamba-3-based hybrid models, we pretrained a 1.5B Mamba-3 SISO hybrid model with four attention layers distributed evenly across the network. We report the downstream evaluation numbers and the real world retrieval performance in the shared response. We would like to highlight that the hybrid model improves the pure Mamba-3 model’s retrieval capabilities significantly, even **outperforming the Transformer baseline by nearly 10 and 7 percentage points on SWDE and FDA** respectively.
>
> --------------
>
> > Could you briefly comment on how the cumulative product of rotation matrices (Π R_i) is handled efficiently during both the parallel training scan and the sequential inference steps?
>
>
> To evaluate the cumulative product of block-diagonal rotation matrices, $\prod_i \mathbf{R}_i$, we use two properties: (i) the product of 2x2 rotation matrices is also a 2x2 rotation matrix whose angle is the sum of the angles of the input matrices, and (ii) the product of block-diagonal matrices is block-diagonal with each block given by the product of the corresponding blocks.
>
> For each index $t$, we keep track of the cumulative sum of rotation angles until index $t$, and rotate $\mathbf{B}, \mathbf{C}$ with the corresponding block-diagonal matrix product. For training and prefill, this cumulative sum of angles is computed in parallel using the RoPE trick. For decode, the same operation is computed sequentially by maintaining the running cumulative angle sum.
>
> --------------
>
> > Could you elaborate on the selection of the MIMO rank r=4? Have you explored the sensitivity of model performance and inference latency to different values of r? Is there a risk of diminishing returns or even performance degradation if r becomes too large?
>
> We would like to highlight the trade-off of various $r$ values. Like we recently added to our manuscript on line 332, increasing $r$ increases the prefill and training time as these stages are generally compute bound, but decode latency stays relatively similar due to its memory bound nature (shown in the table below). For our main results, we selected MIMO rank $r=4$ as a reasonable value given our compute constraints. Since the submission, we have ablated MIMO's performance under $r={2,4,8}$ and **find increasing $r$ continues to improve model performance**. Please refer to our shared response for a more detailed discussion on the impact of $r$ on performance.
>
> | MIMO Rank | Wallclock time (μs) | Memory Bandwidth (GB/s) |
> |----------|----------------------|-------------------------|
> | 1 (SISO) | 239                  | 2286                    |
> | 2        | 255                  | 2156                    |
> | 4        | 260                  | 2145                    |
> | 8        | 283                  | 2030                    |
>
> We find that latency is not very sensitive even as $r$ increases which can be attributed to the underlying decode kernel being highly memory bound. Due to its memory bound nature, the kernel can overlap compute with memory IO, leading to very little increase in wall clock time despite more operations. Even at $r=8$, the kernel is still memory bound.

---

> ### Author Response · Authors · 2025-11-30
> **Response to Reviewer kFu5 (2/2)**
>
> > Can you provide more intuition on why this [trapezoidal discretization and BC bias make the short convolution redundant] is the case? Does the learned size-two convolution from the trapezoidal rule effectively capture the same local information as a dedicated convolution layer?
>
> Our hypothesis is that the BC bias introduces an “implicit” Linear Time-Invariant (LTI) component into the model. Specifically, adding a learned offset to the data-dependent B,C projections, allows the model to run LTI and data-dependent SSMs "simultaneously" [7]. Since LTI-SSMs are equivalent to long convolutions with decay, this provides a mechanism for the model to recover the performance of the explicit short convolution layer. A deeper mechanistic interpretation of BC-bias remains an interesting direction for future work.
>
> --------------------
> ## References
>
> [1] [2504.03624] Nemotron-H: A Family of Accurate and Efficient Hybrid Mamba-Transformer Models
>
> [2] [2408.12570] Jamba-1.5: Hybrid Transformer-Mamba Models at Scale
>
> [3] [2510.26692] Kimi Linear: An Expressive, Efficient Attention Architecture
>
> [4] https://qwen.ai/blog?id=4074cca80393150c248e508aa62983f9cb7d27cd
>
> [5] [2405.21060] Transformers are SSMs: Generalized Models and Efficient Algorithms Through Structured State Space Duality
>
> [6] [2412.06464] Gated Delta Networks: Improving Mamba2 with Delta Rule
>
> [7] [2505.09022] Block-Biased Mamba for Long-Range Sequence Processing

---

### Author Response · Authors · 2025-11-30
**General Response (1/2)**

We thank all reviewers for their time and effort spent reading our paper and providing positive, thoughtful feedback and suggestions. All reviews highlighted the **theoretically grounded SSM-focused improvements** and the thorough **empirical validation across numerous scales which show the strengths of Mamba-3 over current state-of-the-art sub-quadratic models**, e.g., Mamba-2, Gated DeltaNet (Reviewers kFu5, 5Dn9, h5BJ, 2tn7). The main suggestions revolve around exploring hybrid models and their retrieval ability (Reviewers kFu5, h5BJ) and scaling up our MIMO results to larger scales and exploring MIMO design choices, e.g., choice of rank $r=4$ (Reviewers kFu5, h5BK, 2tn7). We are currently running many of these experiments, and report currently finished experiments below. Finished experiments include:

- **Pretrained a 1.5B hybrid Mamba-3 model** composed of 4 self-attention layers with the remaining layers being SISO Mamba-3. We demonstrate that hybrid Mamba-3 models **outperforms all other models at real-world retrieval tasks**, including the Transformer baseline.
- **Pretrained a 820M MIMO Mamba-3 model**, which continues to **outperform the SISO variant by average ~1%** on downstream language modeling tasks.
- **Expanded MIMO rank *r* ablations** that explore performance trade-offs across various *r*.

We are also **currently training other hybrid model variants and a 1.3B MIMO model**, and will update the final camera ready (and here if finished before the discussion period ends) with the results.

------------

## Core Contributions
Our work is anchored on **improving linear-style models through inference-first and SSM-driven perspectives**. While recent works have significantly advanced models under the linear attention and test-time regression framework [1,2,3,4], we show that viewing modeling through a SSM lens can open up a **rich design space** through various perspectives.


- From a **methodological perspective**, we introduced several new techniques----trapezoidal discretization, complex-valued state update, and multi-input, multi-output system----grounded in SSM theory that significantly diverge from other modern linear models, opening up new avenues of improving performance.
- From an **empirical perspective**, we show that Mamba-3 is better than prior most popular / strongest linear models (Mamba-2 and GDN).
- From a **practical perspective**, the SSM lens also leads to faster models (stemming from simple but effective update rules rather than rank-one updates used in many current linear attention models) and easier implementation. The Triton implementation of our Mamba-3 kernels is  on par with those of the public Triton Mamba-2 and Gated Deltanet (GDN).

**We plan to open-source all model code kernels**. In particular, we will release the Triton implementation of our Mamba-3 kernels, which currently achieves speed on par with those of Mamba-2 and Gated Deltanet (GDN). We will also release kernels written in CuTe DSL, a lower-level language compared to Triton, which will improve speed further. We are continuing to optimize both Mamba-2 and Mamba-3 kernels; we have an internal Mamba-2 that is 2-3x faster than the existing kernels, and aim to get Mamba-3 to comparable speed.

------------

## 1.5B Hybrid Mamba-3 Model
We pretrained a 1.5B hybrid Mamba-3 model with a 5:1 ratio of Mamba-3-based to self-attention-based blocks. All mixer layers are followed by an MLP, and we place the self-attention+MLP block at the end of the five Mamba-3+MLP blocks. This interleaved pattern is repeated four times, and the training procedure follows that used for other 1.5B models. We are currently training the other hybrid models (e.g. hybrid GDN) and will include them in the final paper after training concludes.

| Model           | SWDE    | SQD     | FDA     | TQA     | NQ      | DROP    | Avg LM Acc |
|----------------|---------|---------|---------|---------|---------|---------|------------|
| Transformer    | 48.9   | 46.6   | 58.4   | **67.5** | 31.7   | 26.4   | 55.7      |
| Gated Deltanet | 32.7   | 40.0   | 28.3   | 63.5   | 25.7   | 24.5   | 55.7      |
| Mamba 2        | 30.7   | 39.1   | 23.7   | 64.3   | 25.1   | **28.5** | 55.4      |
| Mamba 3        | 28.5   | 40.1   | 23.4   | 64.5   | 26.5   | 27.4   | **56.4**  |
| Mamba 3 Hybrid | **58.5** | **47.0** | **65.9** | 64.8   | **33.4** | 27.0   | **56.4**  |

We highlight that the addition of only a few self-attention layers drastically improves the model’s performance on real-world retrieval, **increasing performance beyond even the Transformer baseline**, while maintaining averaged downstream language modeling (LM) performance comparable to that of pure Mamba-3 SISO.

---

> ### Author Response · Authors · 2025-11-30
> **General Response (2/2)**
>
> ## 820M Mamba-3 MIMO Model
> We pretrained a Mamba-3 MIMO rank $r=4$ at the next largest scale (the original submission reported numbers for a 440M model). We find the 820M MIMO variant **continues to outperform all other baselines** on all but one downstream evaluation task, and **outperforms the SISO variant by nearly an entire percentage point**.
>
>
> | Model         | FW-Edu PPL | LAMB. PPL | LAMB. Acc | HellaS. AccN | PIQA Acc | Arc-E Acc | Arc-C AccN | WinoGr. Acc | OBQA Acc | Avg Acc |
> |--------------|------------|-----------|-----------|--------------|----------|-----------|------------|-------------|----------|---------|
> | Transformer  | 11.42       | 15.0      | 44.7      | 57.2         | 72.6     | 71.6      | 39.2       | 57.7        | 26.8     | 52.8    |
> | Gated DeltaNet | 11.39       | 12.7      | 47.1      | 57.5         | 72.6     | 72.5      | 38.8       | 57.9        | 30.6     | 53.9    |
> | Mamba-2      | 11.35       | 13.8      | 45.0      | 58.1         | 72.5     | 72.3      | 38.7       | 56.8        | 30.2     | 53.4    |
> | Mamba-3      | 11.23       | 12.9      | 47.2      | 58.8         | 73.6     | 72.7      | 40.2       | 58.4        | 30.0     | 54.4    |
> | Mamba-3 MIMO | **11.11**       | **11.8**  | **49.5**  | **59.2**     | **73.7** | **74.7**  | **41.2**   | **59.9**    | 28.6     | **55.3** |
>
>
> ------------
>
>
> ## MIMO Rank r Ablations
> To ablate the effect of MIMO rank $r$ on model performance, we trained 440M Mamba-3 MIMO models across values $r={2,4,8}$ to Chinchilla optimal token count. We maintained the same experimental configurations as those used for the SISO ablations except the doubling of learning rate to simulate larger scale runs [5].  We report the pretraining test perplexity and downstream language modeling performance below.
>
> | Mamba-3 MIMO Rank $r$    | FW-Edu PPL | LAMB. PPL | LAMB. Acc | HellaS. AccN | PIQA Acc | Arc-E Acc | Arc-C AccN | WinoGr. Acc | OBQA Acc | Avg Acc |
> |----------|------------|-----------|-----------|--------------|----------|-----------|------------|-------------|----------|---------|
> | 1 (SISO) | 15.43       | 29.2      | 35.4      | 43.5         | 67.9     | 63.9      | 30.1       | 52.8        | 25.0     | 45.5    |
> | 2        | 15.22       | 27.9      | 35.8      | 43.8         | 68.5     | 62.4      | 30.4       | 52.5        | 24.6     | 45.4    |
> | 4        | 15.12       | 27.3      | 36.5      | 43.9         | 69.2     | 64.4      | 30.3       | 52.7        | 23.8     | 45.8    |
> | 8        | 14.99       | 25.8      | 36.9      | 44.3         | 68.7     | 65.4      | 31.1       | 53.9        | 24.2     | 46.4    |
>
>
>
> ------------
> ## References
>
> [1] [2412.06464] Gated Delta Networks: Improving Mamba2 with Delta Rule
>
> [2] [2506.02475] Comba: Improving Bilinear RNNs with Closed-loop Control
>
> [3] [2407.04620] Learning to (Learn at Test Time): RNNs with Expressive Hidden States
>
> [4] [2506.05233] MesaNet: Sequence Modeling by Locally Optimal Test-Time Training
>
> [5] [2309.14322] Small-scale proxies for large-scale Transformer training instabilities

---

### Meta-Review · Area_Chair_PhAr · 2026-01-06

**Summary:**

This submission introduces Mamba-3, an “inference-first” state-space / linear-time sequence model that aims to improve over prior sub-quadratic backbones (notably Mamba-2 and Gated DeltaNet) along three dimensions: modeling quality, state-tracking capability, and real-world decode efficiency. The core methodological contributions are:

1. Generalized trapezoidal discretization to improve recurrence expressivity/approximation quality vs Euler-style discretizations.
2. Complex-valued dynamics implemented efficiently via an equivalence to data-dependent RoPE-like rotations, enabling stronger state tracking on formal-language tasks.
3. MIMO (multi-input multi-output) update to increase decode arithmetic intensity (mitigating memory-bound decoding) without increasing state size.

The paper provides extensive experiments across multiple model scales (180M–1.5B) and multiple task types (LM, retrieval, formal/state-tracking, latency).

Strengths
1. Principled, well-motivated methodological advances grounded in SSM framing, rather than ad-hoc architectural tweaks.
2. the complex/dynamic-rotation mechanism yields good gains on state-tracking/formal language tasks (a known weakness area for many 3. MIMO is a coherent and practically motivated response to memory-bound decoding
4. multiple scales trained under a common protocol; ablations generally isolate contributions; Pareto-style comparisons align with the “inference-first” story.

Weaknesses

Some reviewers wanted additional linear-family baselines (e.g., block-biased variants, RWKV-like, other retrieval-optimized linear models).

**Reviewer Concerns:**

The rebuttal is substantial and addresses many of the borderline reviewers’ most important concerns:

1. Authors add a hybrid Mamba-3 + attention model demonstrating large retrieval gains. This directly answers the “retrieval gap” criticism in a practical way, while implicitly conceding that pure SSMs remain limited.
2. Authors report larger-scale MIMO (820M) plus rank ablations and latency sensitivity, strengthening the MIMO story beyond a single small point.
3. Authors provide kernel stack details, fusion levels, hardware setup, and prefill+decode comparisons, plus a strong transformer system baseline (vLLM). This materially improves reproducibility and fairness.
4. Authors acknowledge the needed condition for second-order guarantees and move assumptions/proof into an appendix

still outstanding:
Retrieval limitations for pure Mamba-3 remain. The rebuttal largely resolves this by positioning hybrids as the path forward, but the paper should incorporate this framing explicitly (and avoid overclaiming fixed-state retrieval).

**Reviewer Scores:**

h5BJ maye increase to 7, others may stay at current scores

---

### Decision · Program_Chairs · 2026-01-26

Accept (Oral)